

An evaluation of long-term physical and hydrochemical measurements at the Sylt
Roads Marine Observatory (1973-2019), Wadden Sea, North Sea
Johannes J. Rick[1], Mirco Scharfe[3], Tatyana Romanova[1], Justus E.E. van
Beusekom[4], Ragnhild Asmus[1], Harald Asmus[1], Finn Mielck[1], Anja Kamp[5], Rainer
Sieger[3], Karen Helen Wiltshire[1,2]
[1] Alfred-Wegener-Institut, Helmholtz-Zentrum für Polar- und Meeresforschung,
Wattenmeerstation Sylt, Hafenstraße 43, D-25992 List, Germany
[2] Alfred-Wegener-Institut, Helmholtz-Zentrum für Polar- und Meeresforschung,
Meeresstation, Biologische Anstalt Helgoland, Kurpromenade, D-27498 Helgoland,
Germany
[3] † deceased
[4] Helmholtz-Zentrum Hereon, Institute of Carbon Cycles, Department Aquatic
Nutrient Cycles, Max-Planck-Straße 1, 21502 Geesthacht
[5] Hochschule Bremen - City University of Applied Science, Fakultät 5, Abt. Schiffbau
und Meerestechnik, Nautik, Biologie, Bionik, Neustadtswall 30, 28199 Bremen
*Correspondence to:* Johannes J. Rick (jrick@awi.de)



1. Abstract
The Sylt Roads pelagic time series covers physical and hydrochemical
parameters at five neighboring stations in the Sylt-Rømø Bight, Wadden Sea,
North Sea.  Since the beginning of the time series in 1973, sea surface
temperature (SST), salinity, ammonium, nitrite, nitrate and soluble reactive
phosphorus (SRP) were measured twice a week. Other parameters were
introduced later (dissolved silicate (Si) – since 1974, pH - since 1979, dissolved
organic nitrogen (DON) - since 1996, dissolved organic phosphorus (DOP) - since
2001, chlorophyll *a* - since 1979, suspended particulate matter (SPM) - since
1975) and in case of dissolved oxygen were already discontinued (1979-1983). In
the years 1977, 1978 and 1983 no sampling took place. Since the start of the
continuous sampling in 1984, the sea surface temperature in the bight has risen
by +1.11 °C, with the highest increases during the autumn months, while the pH
and salinity decreased by 0.23 and 0.33 units, respectively. Summer and autumn
salinities are generally significantly elevated compared to spring and winter
conditions. Dissolved nutrients (ammonium, nitrite, nitrate and SRP) displayed
periods of intense eutrophication (1973 – 1998) and de-eutrophication since
1999. Silicate showed significantly higher winter levels since 1999. Interestingly,
phytoplankton parameters did not mirror these large changes in nutrient
concentrations, as a seasonal comparison of the two eutrophication periods
showed no significant differences with regard to chlorophyll *a*. This phenomenon
might be triggered by an important switch in nutrient limitation during the time
series: With regard to nutrients, the phytoplankton was probably primarily limited
by silicate until 1998, while since 1999 SRP limitation became increasingly
important.



Repository-Reference: Rick et al. (2017b-e, 2020a-o) and Rick et al. submitted:
doi:10.1594/PANGAEA.150032, 873549, 873545, 873547, 918018, 918032,

50    918027, 918023, 918033, 918028, 918024, 918034, 918029, 918025, 918035,

51    918030, 918026, 918036, 918031

2. Introduction
The Sylt-Rømø Bight (SRB) is a Marine Protected Area (MPA) in the Wadden Sea
UNESCO World Heritage area since 2009. It is a large tidal lagoon (ca. 400 km$^2$) in
the northern part of the Wadden Sea (SE North Sea). In the previous century two
causeways connecting the islands of Rømø and Sylt with the mainland were built.
Since then a narrow inlet between Sylt and Rømø is the only connection with the
open German Bight through which almost 50% of the bights' water is exchanged
each tidal cycle. Local riverine discharge is estimated to be 0.1 % of the total water
input. Tides are semidiurnal with a range of about 2m. At mean low tide 33% of the
bight is exposed, 10% of the remainder comprising deep channels with a maximum
depth of 40m and 57% is a shallow subtidal area with depths less than 5m (Gätje &
Reise, 1998, Figure 1).
In 1973 the Sylt Roads **L**ong **T**erm **E**cological **R**esearch time series (Sylt Roads
LTER) was initiated in this hydrographically and ecologically interesting area. This
consists of a "twice a week" sampling of oceanographic, hydrochemical and
biological (phyto-, zooplankton, fish) parameters. Meanwhile, most of these Sylt
Roads data (> 1000 data sets) has been published online in the open access data
bank PANGAEA (www.pangaea.de).  In this work we summarize for the first time the
information on physical and hydrochemical parameters of this time series and
provide a brief overview of the development over the last 45 years.




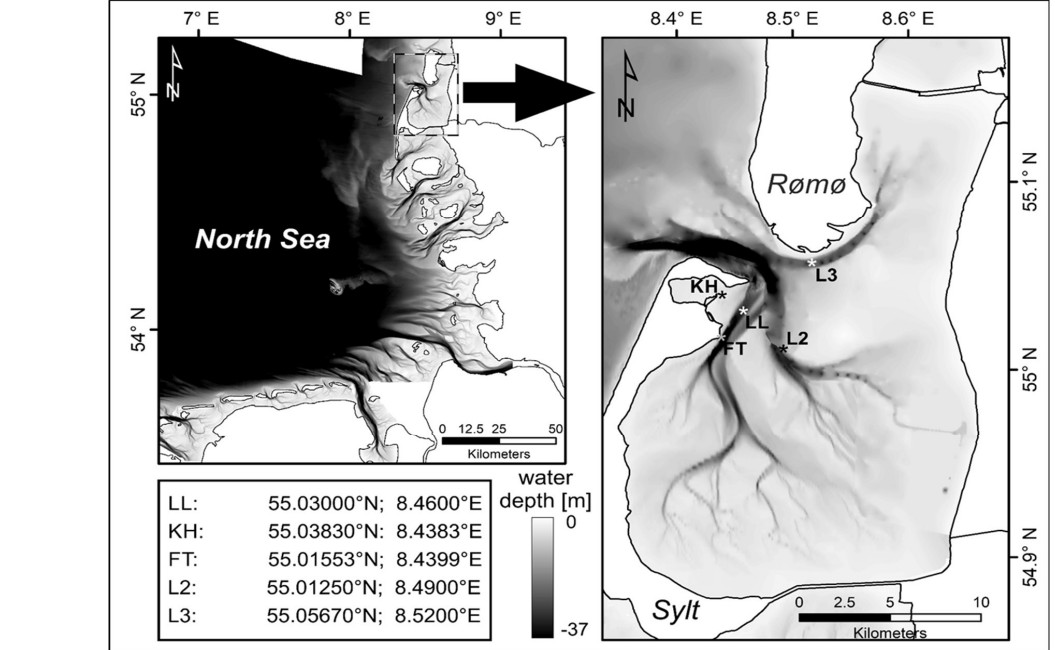

Figure 1: Map of the German Bight with the sampling area (Sylt-Rømø Bight)
enlarged with main sampling stations of the SYLT ROADS LTER time series and
their geographical position. LL: Lister Ley or List Reede, KH: entrance Königshafen,
FT: List Ferry Terminal, L2 and L3: List 2 and 3 stations sampled in early part (until
1991) of the time series only.

## 3.  Data coverage and parameters measured
Coverage:
North:  55.01250 - 55.05670; East: 8.43830 - 8.52000

Location names and positions:
LL: List_Reede (Lister_Ley), Sylt Rømø Bight, Wadden Sea, North Sea: North: 55.03000;
East: 8.46000
L2: List_2, Sylt-Rømø Bight, German Bight Wadden Sea, North Sea: North:  55.01250; East:
8.49000

L3: List_3, Sylt-Rømø Bight, German Bight Wadden Sea, North Sea: North: 55.05670; East:

8.52000

KH: List_Entrance_Königshafen, Sylt-Rømø Bight, German Bight Wadden Sea,



North Sea: North 55.03830; East: 8.43830

FT: List_Ferry_Terminal, Sylt-Rømø Bight, German Bight Wadden Sea, North Sea:

North: 55.01553; East: 8.43990

Date/Time Start: 1973-06-28T00:00:00

Date/Time End:   2019-12-31T00:00:00


| Parameter | Short Name | Unit | Comment |
|---|---|---|---|
| DATE/TIME | Date/Time | | Geocode |
| DEPTH, water | Depth water | M | Geocode |
| Salinity | Sal | | |
| Temperature, water | Temp | °C | |
| pH | pH | | |
| Dissolved Oxygen | $O_2$ | µmol/l | |
| Chlorophyll *a* | Chl *a* | µg/l | Filtered through GFC, stored frozen (-20°C), Extraction by Acetone |
| Phosphate | $[PO_4]^{3-}$ | µmol/l | Filtered 0.4 µm Nucleopore, stored frozen (-20°C) |
| Silicate | $Si(OH)_4$ | µmol/l | Filtered 0.4 µm Nucleopore, stored frozen (-20°C) |
| Ammonium | $[NH_4]+$ | µmol/l | Filtered 0.4 µm Nucleopore, stored frozen (-20°C) |
| Nitrite | $[NO_2]^-$ | µmol/l | Filtered 0.4 µm Nucleopore, stored frozen (-20°C) |
| Nitrate | $[NO_3]^-$ | µmol/l | Filtered 0.4 µm Nucleopore, stored frozen (-20°C) |
| Nitrogen, organic, dissolved | DON | µmol/l | Filtered precombusted GFC, stored frozen (-20°C) |
| Phosphorus, organic, dissolved | DOP | µmol/l | Filtered precombusted GFC, stored frozen (-20°C) |
| Suspended matter | SPM | mg/l | Filtered 0.4 µm Nucleopore, stored frozen, dried (60°C) |


4.  Instrumentation and methods
Sea surface temperature (SST), salinity, ammonium ($NH_4^+$), nitrite ($NO_2^-$), nitrate
($NO_3^-$), soluble reactive phosphorus (SRP) and reactive silicate (Si) measurements
were started in 1973 and interrupted temporarily in the years 1977, 1978 and 1983.
Temperatures of the sea surface (SST) were gathered using reversing thermometers
(Thomas & Dorey, 1967). For the period 1973 – 1982 the inductive salinometer
method was used for salinity measurements (Brown & Hamon, 1961).  Since 1983,
we measured the salinity using a Guildeline AutoSal 8400B salinometer (Kawano,
2010). pH-measurements were initiated in 1979. Until 1984, diverse pH meters were



applied and since 1985 a WTW pH 3000 Meter is in use. Dissolved oxygen was
measured only during the period from 1979-1983 using the Winkler method (e.g.
Culberson et al., 1991). Table 1 gives an overview on the methods applied within the
time series for several chemical analyses on nutrient components and chlorophyll *a*.
For both DON and DOP filtration we used precombusted CFC filters and filtrates
were frozen at -20°C, while for chlorophyll *a* analysis untreated GFC filters were
employed instead. For gravimetric suspended matter (SPM) analyses we used
precombusted CFC filters from 1975 to 1998, since 1999 0.4 – 0.45 µm
NUCLEOPORE filters were employed.

| parameter | time period | analysis |
|---|---|---|
| soluble reactive phosphate (SRP) | 1973-1983 | Koroleff (1976a) |
| reactive Si (Si) | 1974-1982 | Koroleff (1976b) |
| ammonium ($NH_4^+$) | 1973-1982 | Grasshoff & Johannsen (1972) |
| nitrite ($NO_2^-$) | 1973-1982 | Bendschneider & Robinson (1952) |
| nitrate ($NO_3^-$) | 1973-1982 | Grasshoff & Wenck (1983) |
| SRP, Si, $NH_4^+$, $NO_2^-$, $NO_3^-$ | 1984-ongoing | Grasshoff et al. (1983) |
| dissolved organic nitrogen (DON) | 1996-ongoing | Grasshoff et al. (1983) |
| dissolved organic phosphorus (DOP) | 2001-ongoing | Grasshoff et al. (1983) |
| chlorophyll a (Chl *a*) | 1979-ongoing | Jeffrey & Humphrey (1975) |

Table 1: Compilation of methods applied in the Sylt Roads time series

Since the start of the Sylt Roads time series, six analysts have been engaged in the
hydrochemical analyses (Table 2).

| analyst | time period | years, months |
|---|---|---|
| 1 | 1973 – 09/1977 | 4y 9 m |
| 2 | 10/1978 – 01/1992 | 13y 4m |
| 3 | 09/1992 – 08/1994 | 1y 11m |
| 4 | 10/1994 – 02/1999 | 4y 5m |
| 5 | 05/1999 – 12/2000 | 1y 7m |
| 6 | since 05/2001 | >18y |


Table 2: Analysts within the Sylt Roads hydrochemistry time series



Sampling was mostly conducted from small research vessels (RV Mya till 2012, since
2013 RV Mya II), or sometimes, in severe weather conditions it was land-based at
the List Ferry Terminal. Figure 1 provides an overview on the geographical position
of the main sampling locations in the Sylt-Rømø Bight (SRB).

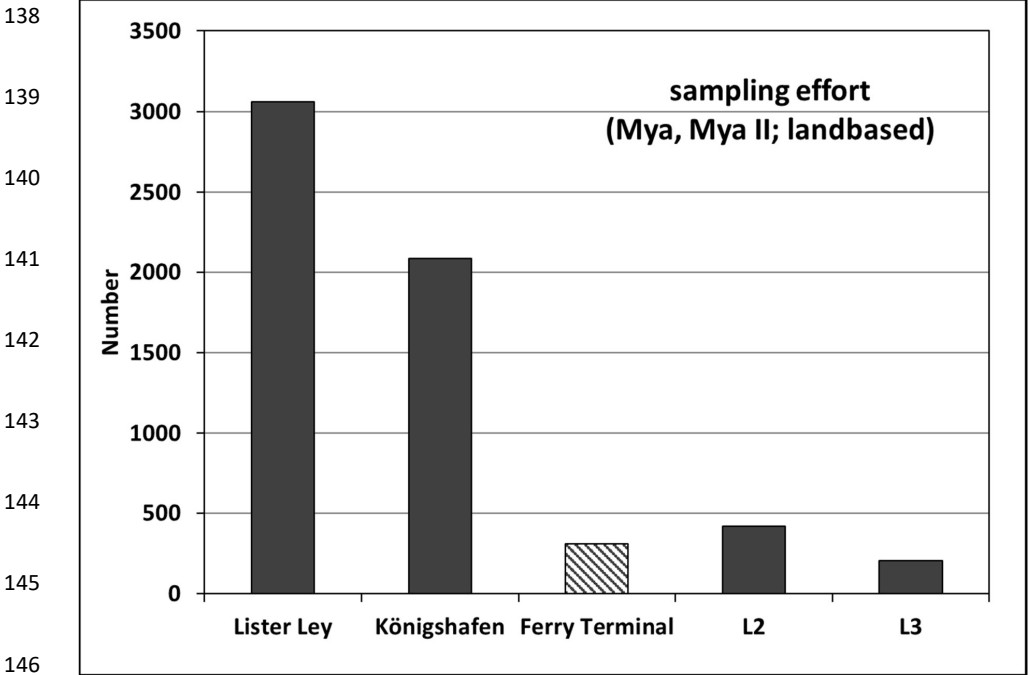

Figure 2a: Overall sampling efforts (ship- and land-based [Ferry Terminal]
campaigns) at the five sampling stations (1973-2019)

5.  Datasets and Discussion
5.1 General description of the basic data
Ship- and land-based sampling efforts are displayed in Figure 2a. The Lister Ley
station (LL) and the Königshafen station (KH) were visited most frequently, while
stations List 2 and 3 (L2, L3) were sampled only during the early periods (1973-1976;
1987-1991) of the time series. Since 1999 the List Ferry Terminal station (FT) was



used as a backup when ship-based sampling was not possible due to adverse
weather conditions. Overall, more than 63.000 data were collected during more than
5.700 RV Mya and Mya II cruises and about 300 land-based sampling efforts at the
List Ferry Terminal. Figure 2b provides an overview of the seasonal sampling efforts
summarized for all stations. Generally, the number of samples per season varied
during the first part of the time series, since 1999 seasonal sampling was more
homogenous. The inserted box plots compare the earlier with the more recent parts
of the time series. For winter and summer sampling significant differences in
sampling effort are obvious (Figure 2b, Table A1 I).

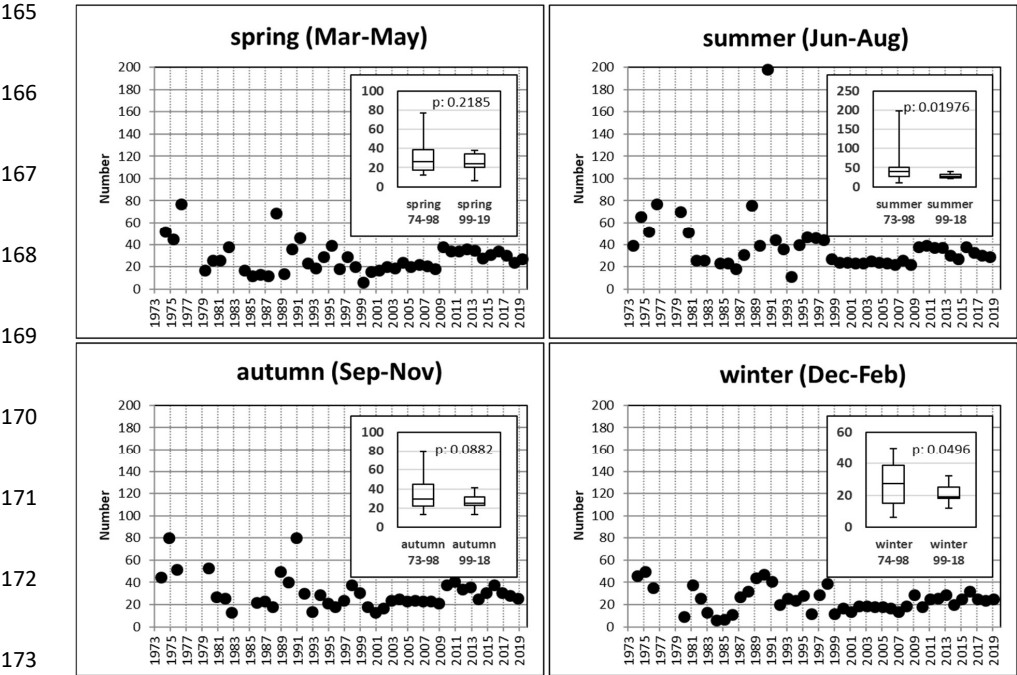

Figure 2b: Seasonal sampling efforts summarized for all Sylt Roads stations in the
SRB (1973-2019). The inserts compare seasonal efforts from early days (1973/74 –
1998) with the more recent part (1999-2019) of the time series.

Most of the measured parameters are shown as original data in Figures 3a-j.  Due to
the physical proximity of stations and the extremely well-mixed waters in the SRB,
data from all sampling stations (Figure 1) were included in the graphs. Most of the
parameters, even salinity (Figure 3a), show seasonal signals.

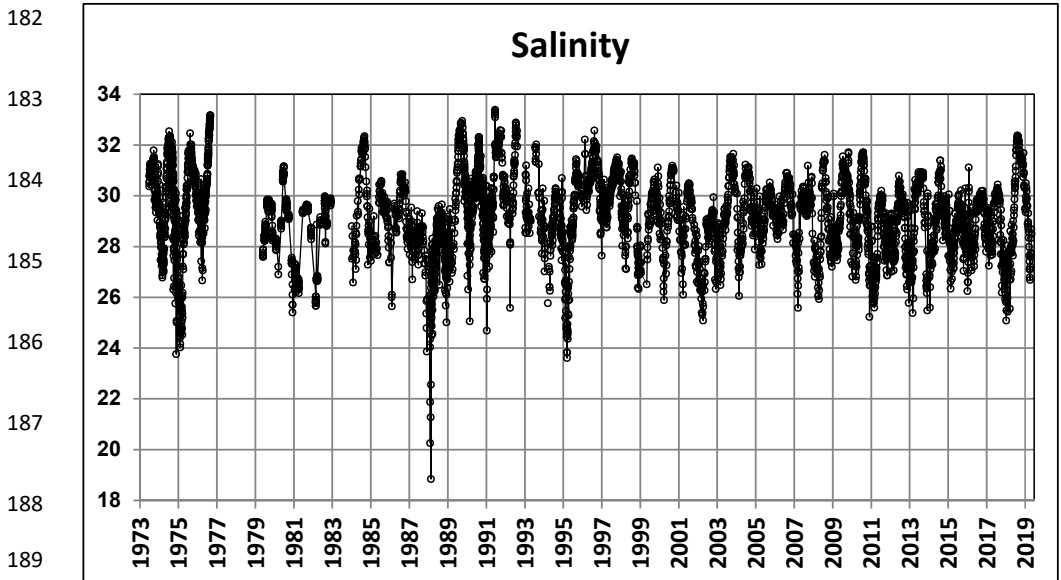

Figure 3a: Salinity time series at Sylt Roads. Data of the five sampling stations
(Figure 1) are included in all subgraphs of Figure 3.

For salinity this is mainly triggered by the enhanced freshwater runoff in late winter
and spring. Seasonal patterns are most evident for the SST (Figure 3b) and the
associated oxygen content of the waters (data not shown) as well as for the major
inorganic nutrients as $NH_4^+$, $NO_2^-$, $NO_3^-$, SRP and reactive silicate (Figures 3c-g). Not
too much should be read into the nutrient data from early years since some (e.g.
$NH_4^+$, SRP) show quite high variability or exceptionally low values (Si, $NO_3^-$)
especially in the initial period (1973-75). From 1992 to 1994 all $NH_4^+$ numbers were
also exceptionally low, which coincided with a specific analyst (Table 2) and are
obviously erroneous. All questionable values were eliminated from the graph (Figure
3c).

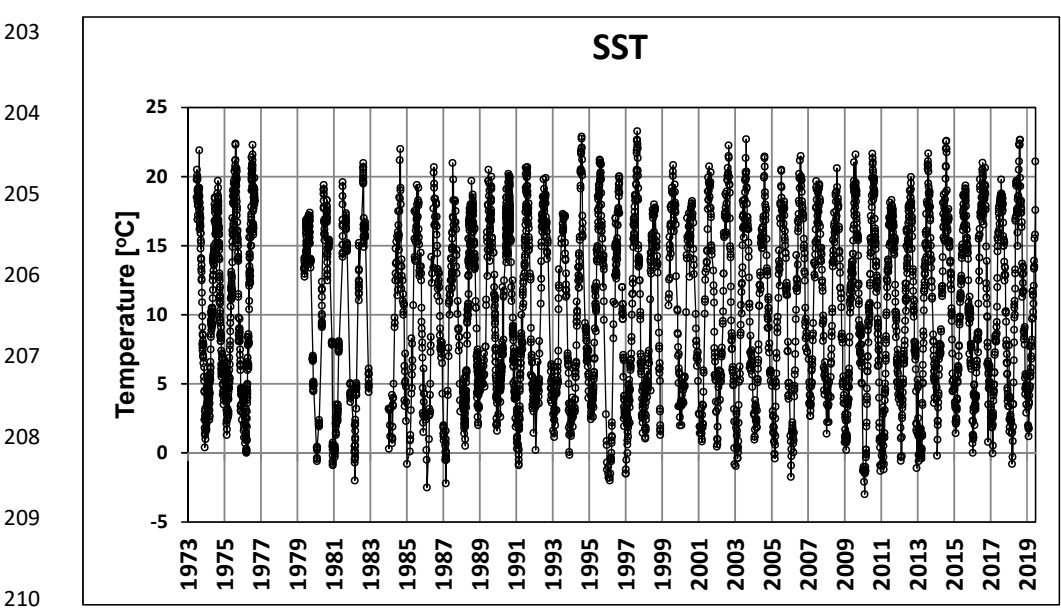

Figure 3b: Time series of the sea surface temperature (SST) at Sylt Roads
Dissolved inorganic nutrients display an opposite behavior compared to the SST with
high values in winter/early spring and minimal numbers during summer. As expected
Chlorophyll *a*, pH (Figure 3h, i) as well as dissolved organic nutrients (data not
shown) are inversely related to levels of inorganic nutrients due to the nutrient uptake
by the phytoplankton.
High SPM is mostly found in winter due to the large amounts of sediment mixed into
the water column by wind forcing (Figure 3j, Bayerl et al., 1998). In summer SPM
decreases to minimum values. A deviation from this pattern was seen in the period
from 1993-1997, which is likely due to inaccurate sample treatment: following the
filtration process, the sea salt retained by the filter material is normally leached out
using distilled water. When the salt is not completely removed in this process the



measured SPM load will be biased. This was probably the case for the 1993-1998
SPM measurements and the respective data should not be used and consequently
have been omitted from the graph.

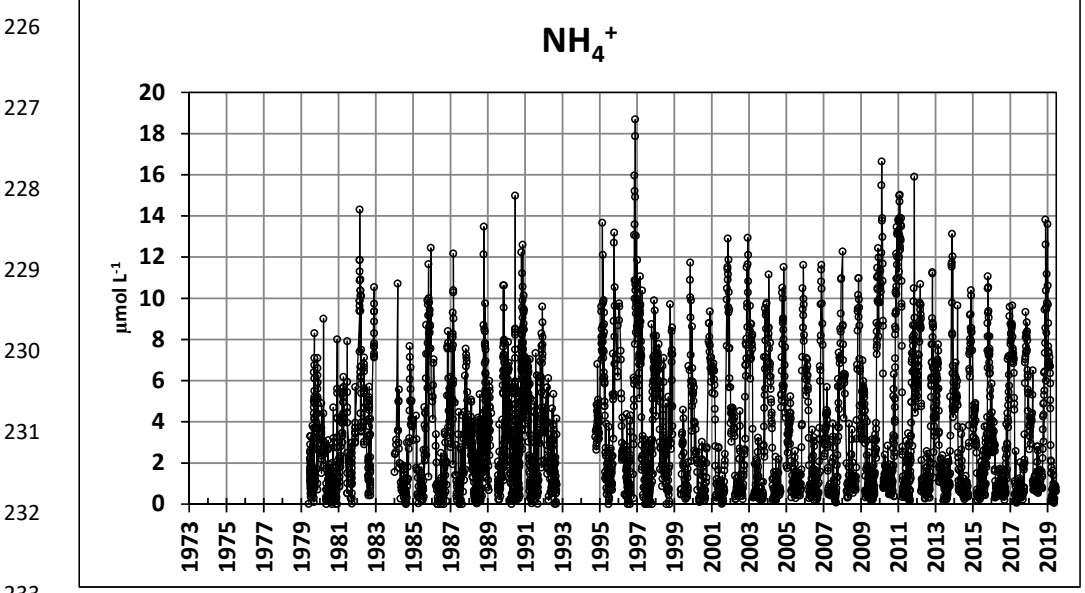

Figure 3c: Development of ammonium concentrations at Sylt Roads (1979-2019).
Data from 1973 - 1978 and 1993 - 1994 were biased and are not shown in the graph.

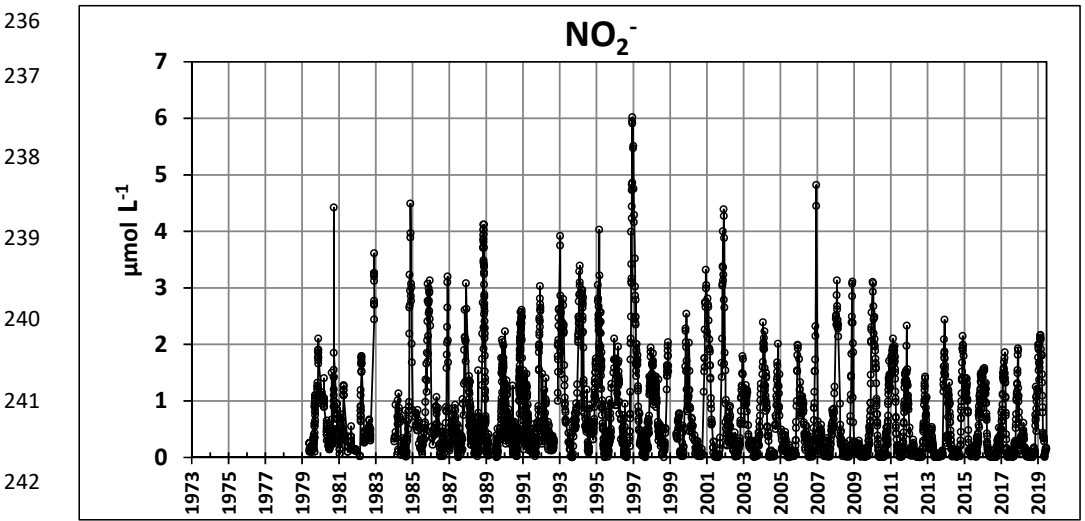

Figure 3d: Development of nitrite concentrations at Sylt Roads.

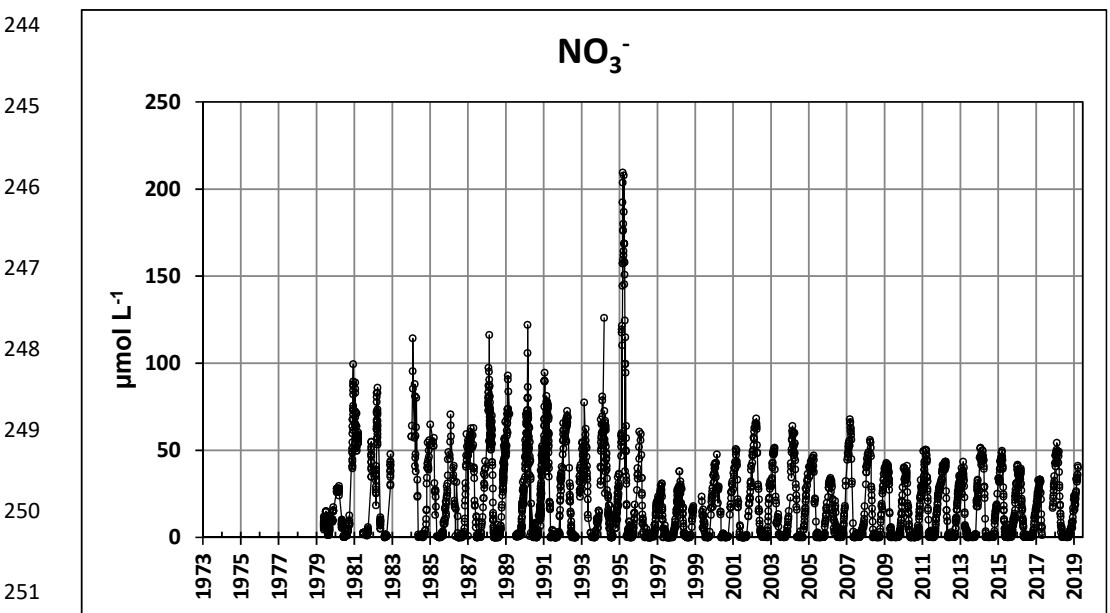

Figure 3e: Development of nitrate concentrations at Sylt Roads.

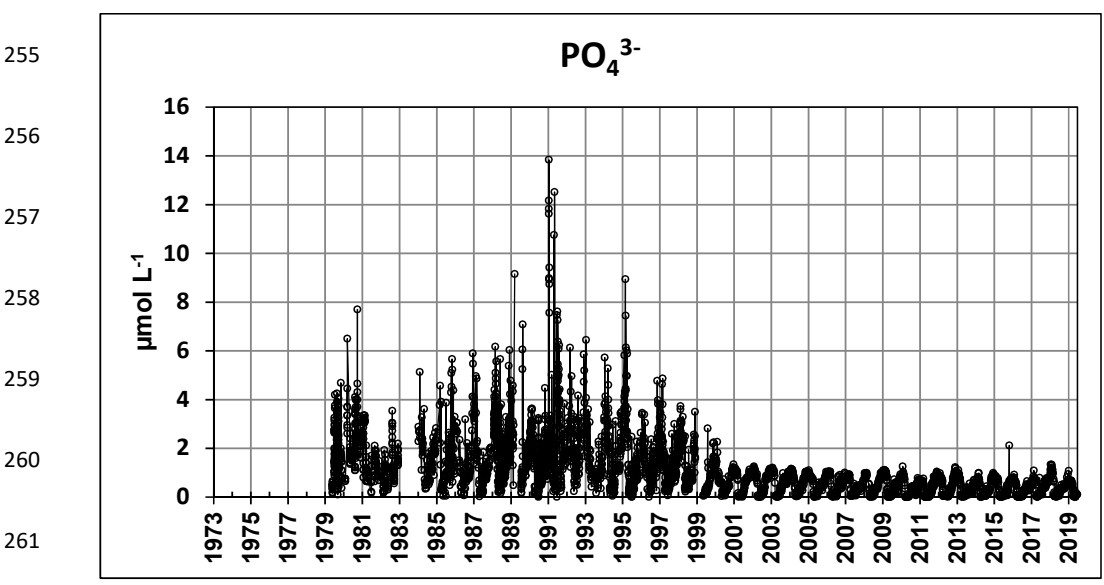

Figure 3f: Development of soluble reactive phosphate (SRP)

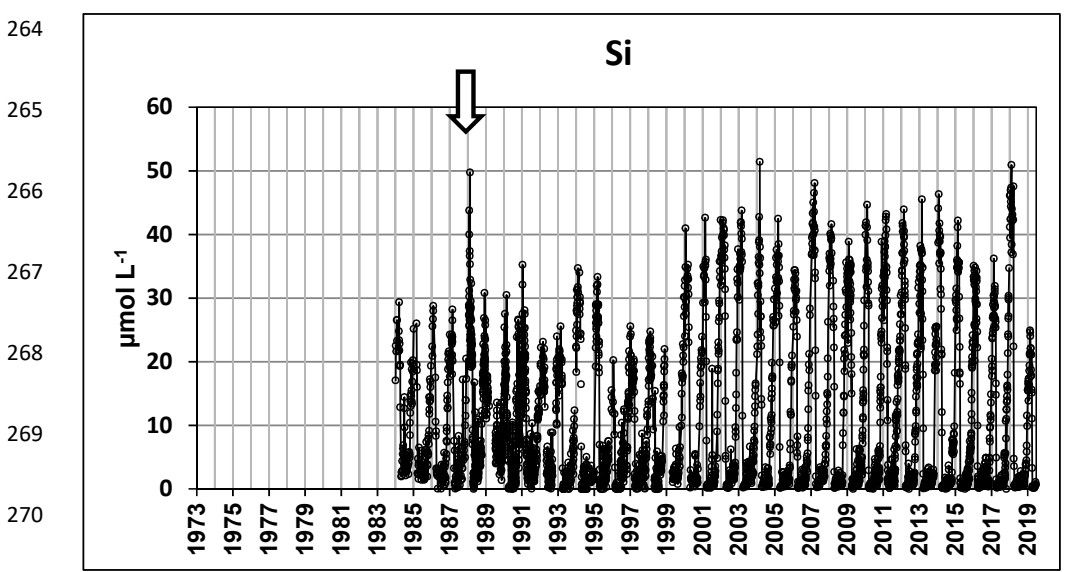

Figure 3g: Development of reactive silicate (Si) concentrations at Sylt Roads. The "1988 Si anomaly" is marked with an arrow

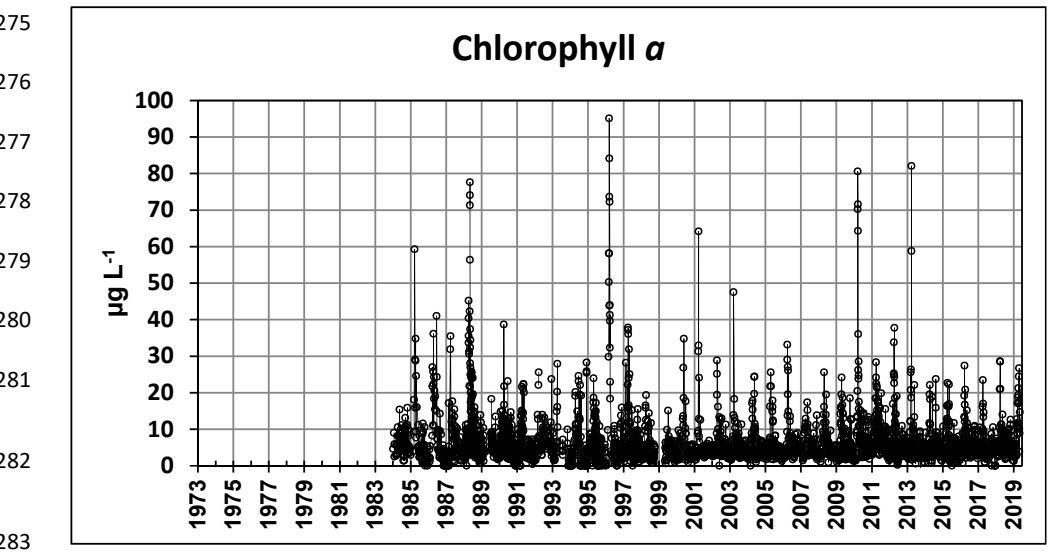

Figure 3h: Development of Chlorophyll *a* concentration at Sylt Roads.

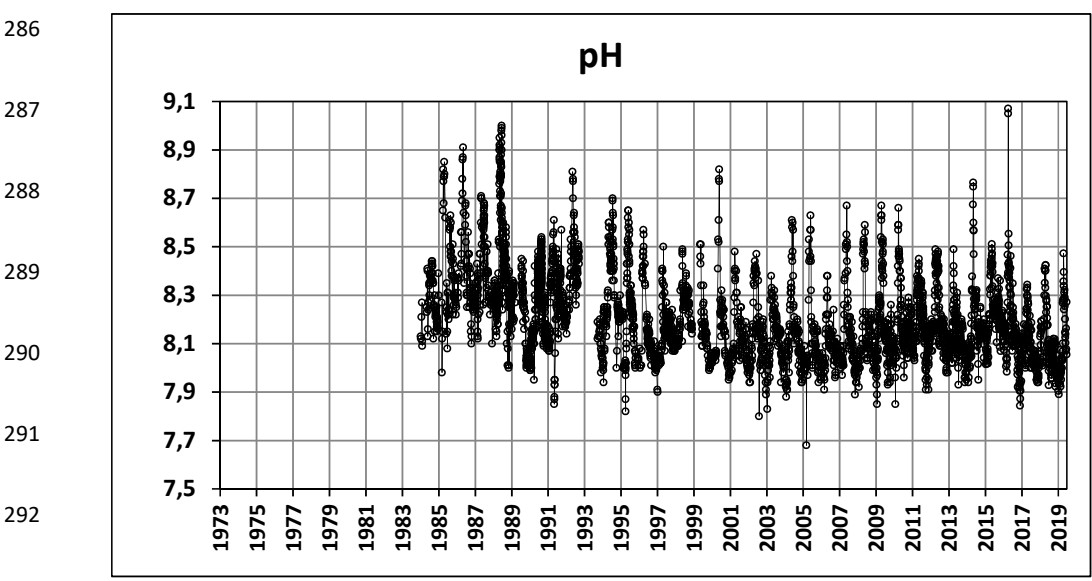

Figure 3i: pH development at Sylt Roads. Data before 1984 and from 1992 were
biased and are not included in the graph.

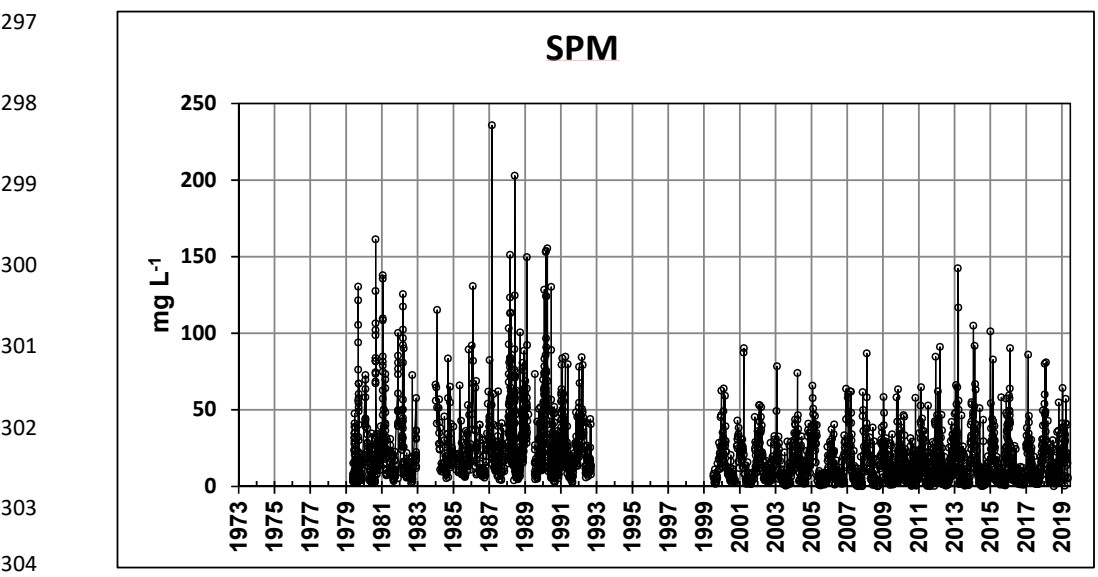

Figure 3j: Development of suspended particulate matter concentrations (SPM) at Sylt
Roads. No data are shown for the period of biased handling (1993-98)



The nutrient plots (e.g. Figure 3e, f) indicate a change in the eutrophication status of
the bight. Until 1998, nitrate as well as SRP concentrations were high, since 1999
they have been decreasing. This is in line with several observations from the
southern North Sea area and mainly due to strong reductions of phosphorus and
nitrogen loads in the rivers Rhine, Ems, Weser and Elbe (e.g. Carstensen et al.,
2006; van Beusekom et al., 2005, 2018, 2019).
Much a higher variability in nutrient values was evident for the high eutrophication
period (1973-1998) compared with more recent times (1999 – 2019) of reduced
nutrient loads. This high variability might be partly related to the fact that till 1998 only
unfiltered nutrient samples were analyzed, from 1999 on the samples were finally
filtered (van Beusekom et al., 2009). The early eutrophication period was additionally
characterized by intense marine or inshore construction and dredging activities.
Sediments originating from the Sylt-Rømø Bight were intensively used for dike
building (e.g. the polders Margarethenkoog and Rickelsbüller Koog), the Hoyer lock
was constructed, the Ruttebüll Lake dredged out and the river Vidà renatured. All
these activities certainly have influenced e.g. the loads of SRP and contributed
potentially to the high variability in nutrient concentrations. An intense blue mussel
fishery in the early period of the time series with its associated dredging impact as
well as the shutdown of the List sewage plant in 2005 might have played an
important role in nutrient variability, too.

5.2 Nutrients, chlorophyll *a*, nutrient ratios and SPM
Since most of the parameters show seasonal signals, it was considered appropriate
to focus on changes for the four main seasons in the course of the time series.
Figure 4a gives an example for the nutrient SRP. For each year in the time series

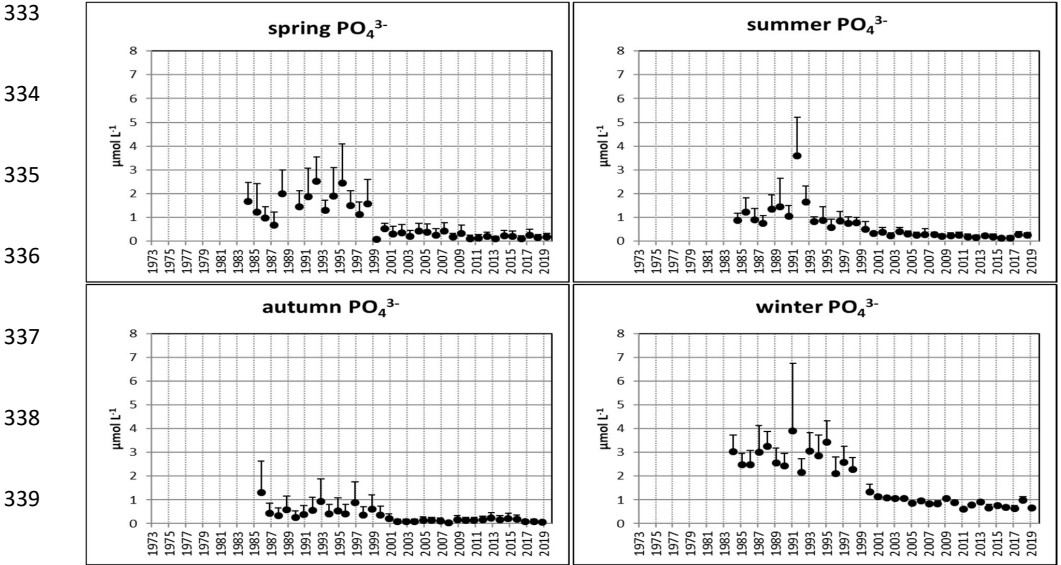

Figure 4a: Development of SRP over the course of continuous measurements (1984-2019) within the Sylt Roads LTER time series. Seasonal averages (Dec, Jan, Feb – winter; Mar, Apr, May – spring; Jun, Jul, Aug – summer; Sep, Oct, Nov – autumn) are displayed with standard error of means (SEM) as error bars.

seasonal averages are presented together with their respective standard errors. As

already seen to some extent in Figure 3f, a first period (1984-1998) of relatively high

values shifts towards a second one (1999-2019) with a lot lower SRP concentrations.

A comparison of both periods using a t-test (two-sided, different variances assumed)

results in highly significantly lower (p: 0.0003 – $1.1 \times 10^{-10}$) and much less variable

SRP values for all seasons in the period of low eutrophication (1999-2019; Figure 4b,

Table A1 a).

Dissolved inorganic nitrogen (DIN, i.e. sum of nitrate, nitrite and ammonium) shows a

similar pattern although the respective t-tests yielded significant differences for spring

(p: 0.017) and winter (p: 0.001) seasons only (Figure 5, Table A1 b).

Silicate (Si), a nutrient important for diatoms, shows a completely different pattern

(Figure 6, Table A1 c). The more recent (1999-2019) low eutrophication winters and

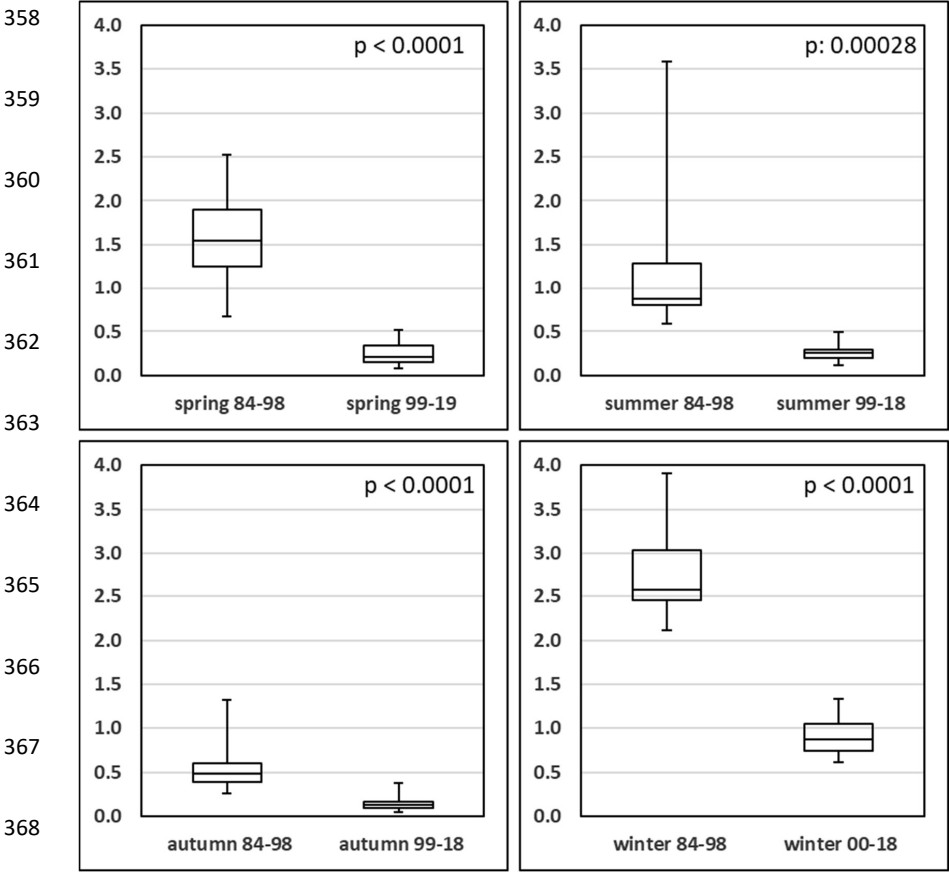

Figure 4b: Seasonal comparison of SPR concentrations [µmol*l$^{-1}$] for high/low
eutrophication periods. Boxplots give median values, with quartiles 1 and 3 attached
as boxes and min and max values shown as endpoints of the error bars. All data
including possible outliers are shown in the graph. The p-values of the respective t-
tests are given in the upper right.

autumns (N and P) showed significantly (p: 1.16 x 10$^{-6}$ and 0.026) elevated Si values
compared with the respective data of high eutrophication (1973-1998). For the spring
comparison Si values remained in the same range. In summer (p = 0.001), the low
eutrophication set showed a significantly lower value. Generally, the variability of Si
was a lot higher in the period from 1973-1998 compared to 1999-2019 (Figure 6;
Table A1 c). Interestingly, the silicate anomaly from 1988 (Raabe & Wiltshire, 2009)
shows its imprint (highlighted in Figure 3g) in the Sylt Roads data, too.

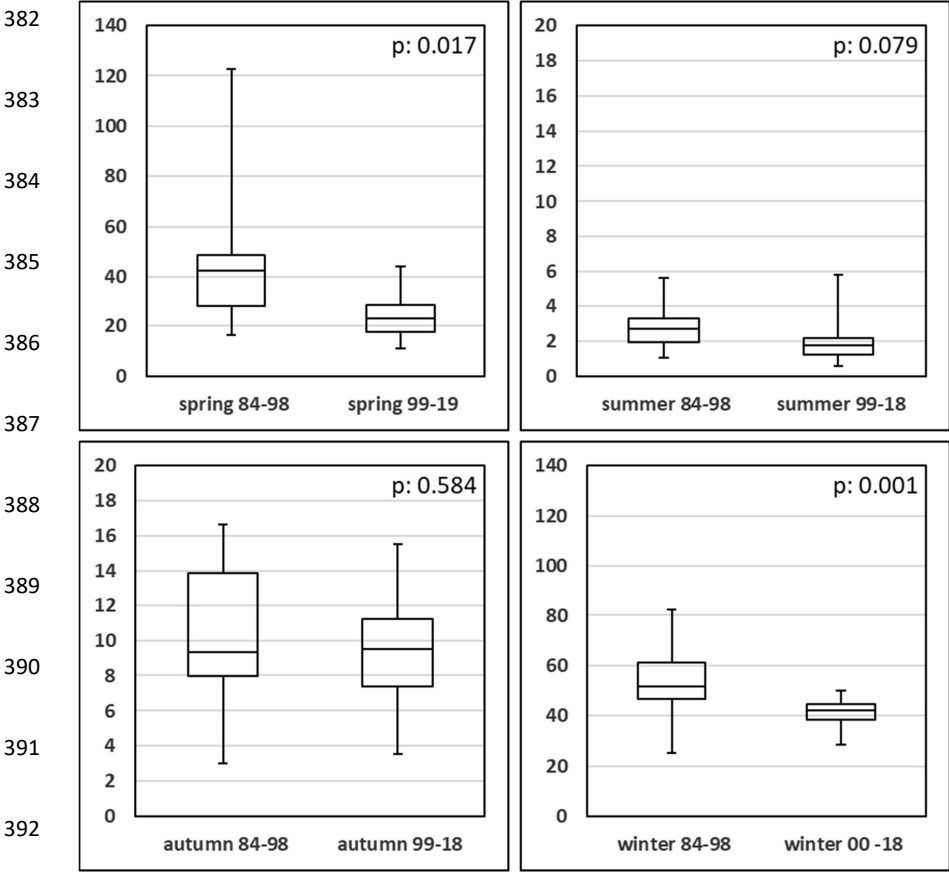

Figure 5: Seasonal comparison (boxplots and t-test p-values) of dissolved inorganic
nitrogen (DIN) concentrations [µmol*l$^{-1}$] for high/low eutrophication periods. Detailed
information is available in Figure 4b.
Despite these large changes in nutrient concentrations, phytoplankton parameters
such as chlorophyll *a* (Figure 3h, 7 and Table A1 i) or phytoplankton carbon (Rick et
al., 2017a) did not shift accordingly, as probably expected (e.g. Cadee & Hegeman,

2002).

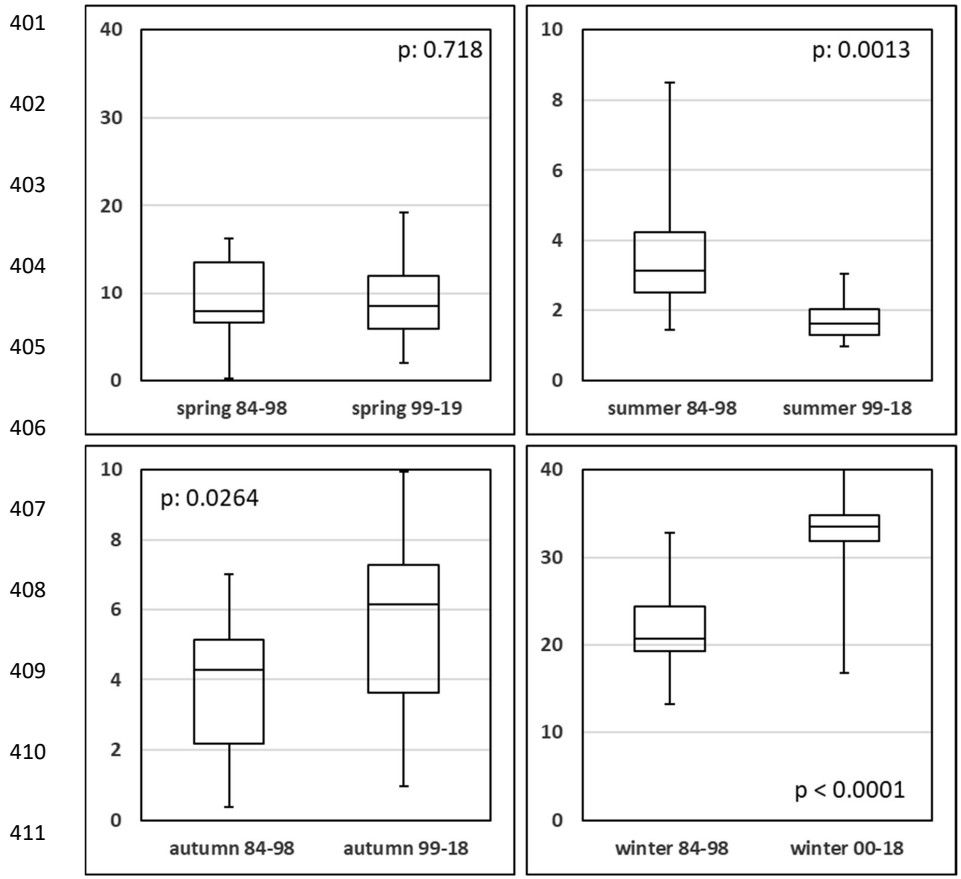

Figure 6: Seasonal comparison of reactive silicate concentrations [μmol*l$^{-1}$] for high/low eutrophication periods.

Planktonic algae are not solely influenced by the total concentrations of single nutrients – but rather it is the nutrient ratios have an essential impact (Dugdale, 1967). For most algae the DIN/SRP ratio (Figure 8, Table A1 j) is of major importance (Redfield, 1934, 1958), diatoms are additionally affected by the DIN/Si (Figure 9, Table A1 k) ratio (Brzezinski, 1985). In Figures 8 and 9 the optimal nutrient ratios, based on molar concentrations, are highlighted as grey bars.

Generally, the DIN/SRP ratio in most cases is highly significantly elevated in the low

eutrophication period when compared with the high eutrophication period (Figure 8).

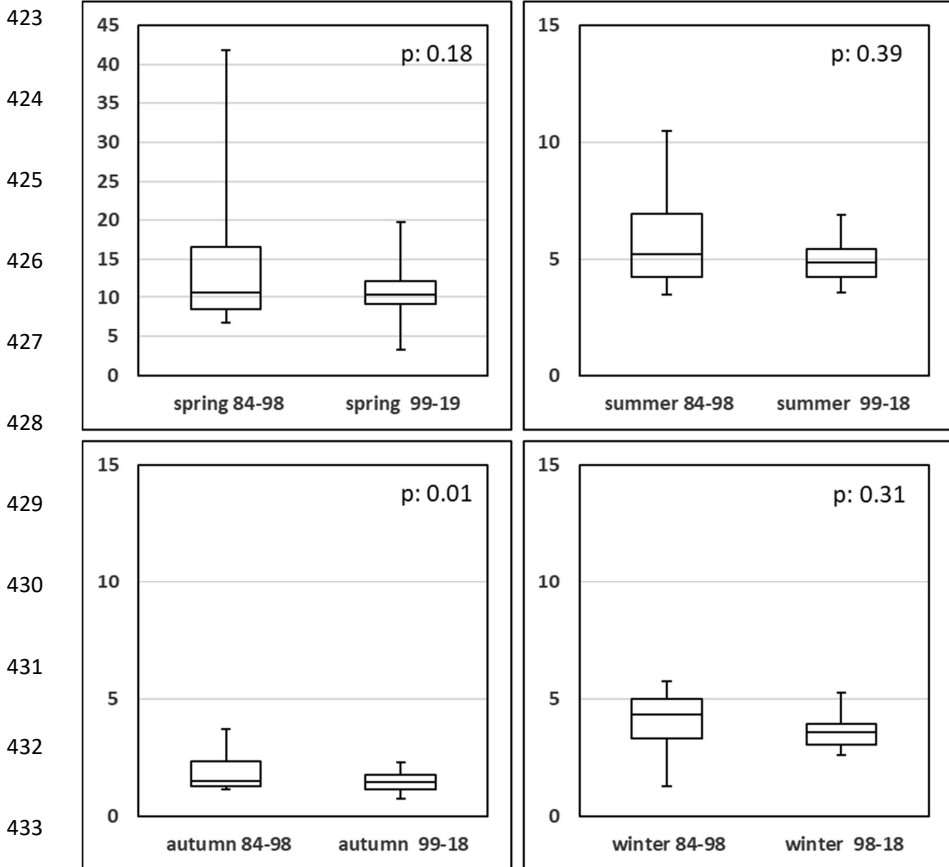

Figure 7: Seasonal comparison of Chlorophyll *a* concentration [µg*l$^{-1}$] for high/low
eutrophication periods.

For winter and spring this change moved the ratio towards an increasing
phosphorous limitation, while for summer and autumn it diminished the N-limitation
during the high eutrophication period.
The spring and winter DIN/Si ratios (Figure 9, Table A1 k) moved from higher (1973-
1998) to more balanced values (1999-2019). For winter (p = 0.018) this change is
significant.  For the summer and autumn comparisons DIN/Si remained close to a
ratio of 1.

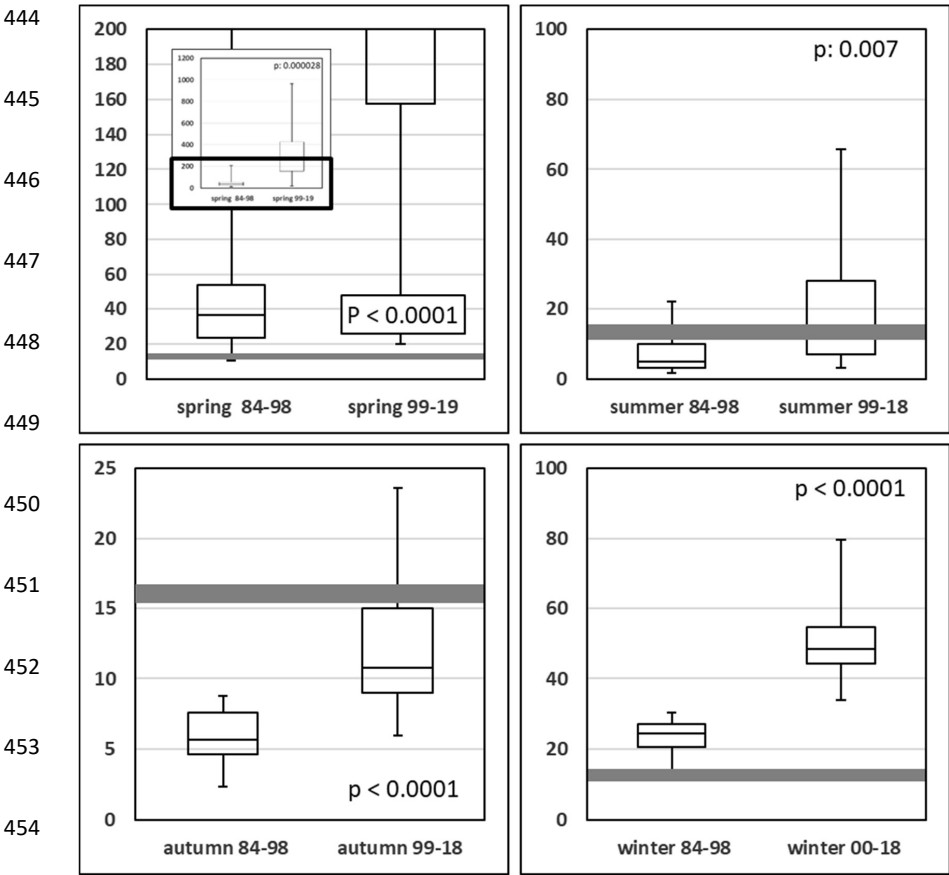

Figure 8: Seasonal comparison of DIN/SRP molar ratios for high/low eutrophication periods. The optimum value around 16 is highlighted as a grey bar. The black boxed part of the spring plot is shown enlarged.

Diatoms are the most prominent phytoplankton group in the bight during all seasons
(Rick & Wiltshire, 2016; Rick et al., 2017a, 2018). In addition to diatoms, solely the
prymnesiophyte *Phaeocystis globosa* (Scherffel, 1899) may add substantially to the
photosynthetic biomass in late spring and early summer (Rick et al., 2017a). During
the period of high phosphorus and nitrogen loads (1973-1998), silicate was probably
not available in sufficient amounts with the result that the diatoms were, at least for
the spring bloom, limited by silicate. Since the decline of SRP and DIN in the second
half of the time series (1999-2019) silicate limitation was replaced by a limitation by

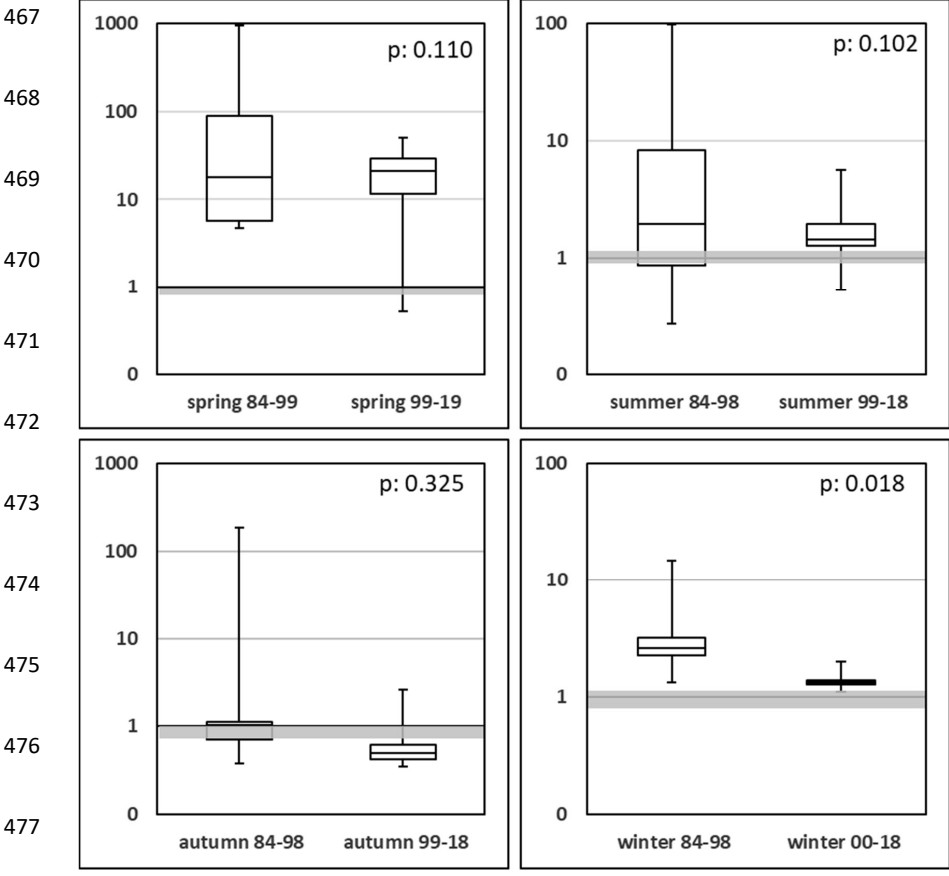

Figure 9: Seasonal comparison of DIN/Si molar ratios for high/low eutrophication periods. The optimum value around 1 is highlighted as light grey bars. Note the log scaled y-axes.

phosphorus. This explains the almost unchanged Chlorophyll *a* pattern despite the
strong nutrient changes (Figure 7, Table A1 i). These results are in accordance with
the findings of Loebl et al. (2009), who studied patterns of phytoplankton limitation
along the southern North Sea coast for the period 1990 to 2005. The authors



concluded that aside from underwater light, silicate limitation of the phytoplankton
was most common followed by the restraining effects of low phosphorus
concentrations.

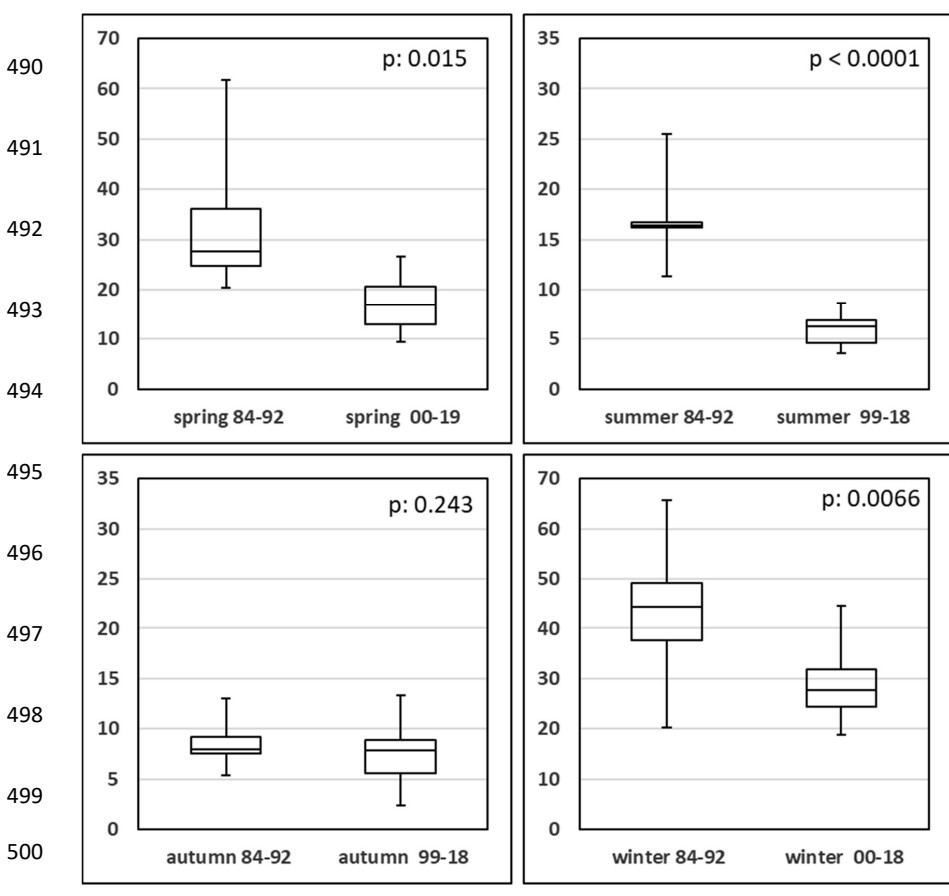

Figure 10:  Seasonal comparison of SPM values [mg*l⁻¹] for high/low eutrophication
periods.

A comparison of seasonal SPM data for both eutrophication periods is given in
Figure 10 and Table A1 h.  Despite the omission of the biased values (1993-1997) a



t-test comparison for all seasons resulted in significantly lower values for the low
eutrophication period (1999-2019). This cannot be explained either by lowered
plankton biomass (Rick et al., 2017a) or by less sediment input into the water during
these years. We assume a change in the SPM methodology might be the cause.
Since 1999, Nucleopore filters were used instead of GF/C-filters. Therefore,
comparisons of recent and earlier SPM data should be avoided.
5.3 Development of sea surface temperature, salinity and pH
SST rose since the start of continuous measurements in 1984 until 2019 by 1.11 °C,
which is close to the temperature development at Helgoland Roads (Wiltshire &
Manly, 2004). Summers warmed by 1.24 °C, spring seasons by 1.14 °C, autumn
seasons actually by 2.04 °C but winters even cooled slightly by -0.16 °C (Figure 11a).

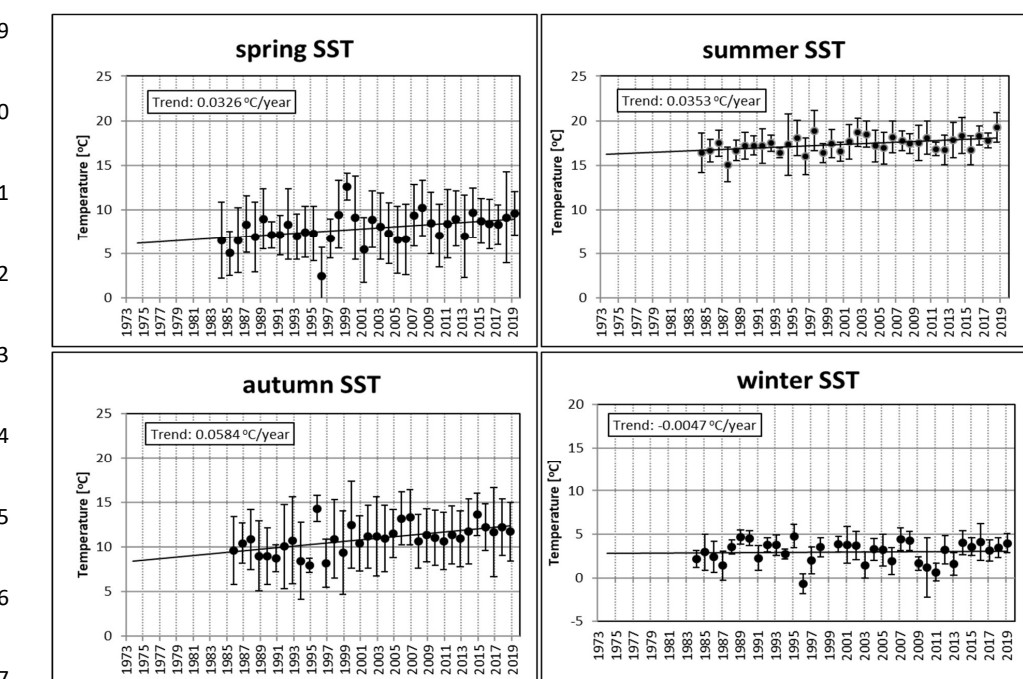

Figure 11a: Development of SST over the course of the Sylt Roads LTER time
series. Seasonal averages with standard error of means (SEM) as error bars. Data
on linear seasonal trends (1984-2019) are shown in boxes






Figure 11b and Table A1 d show a t-test comparison of identical seasons for the two
periods defined in the previous chapter. For all seasons the period 1999-2019 shows
higher average SST values compared with the earlier years of the time series. This
finding is significant for summer (p: 0.043) and autumn data (p: 0.0004).

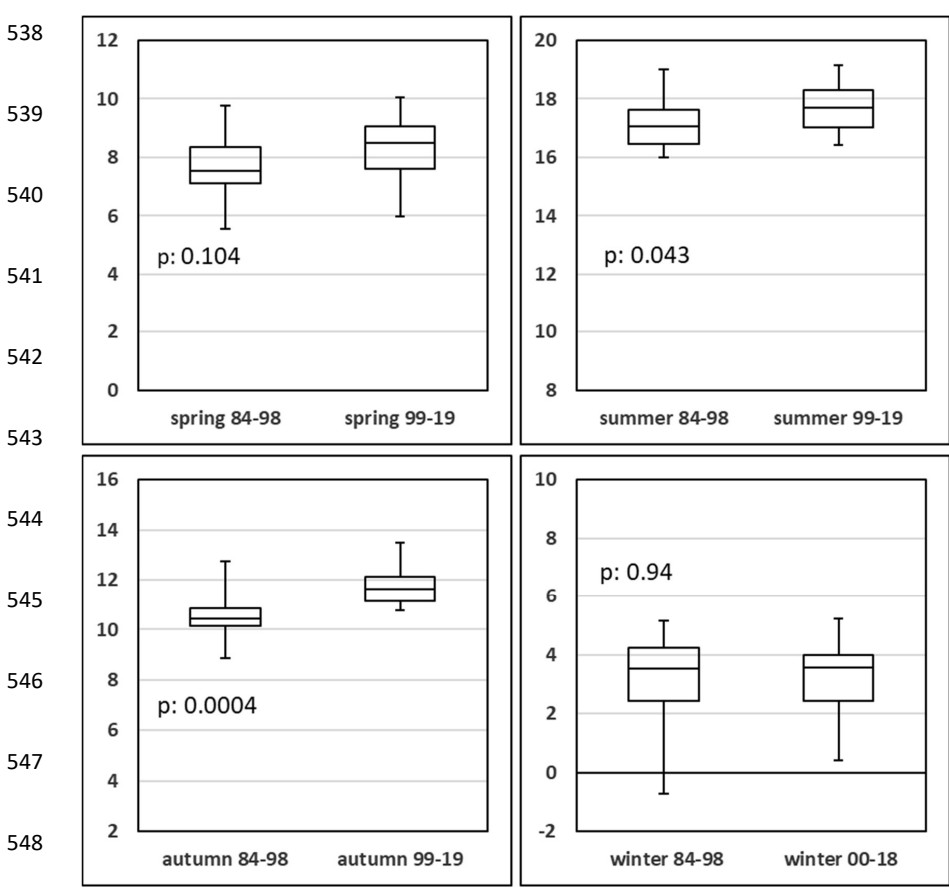

Figure 11b: Seasonal comparison of SST values [°C] for the early and recent part of
the time series.

Figure 12 and Table A1 f give an "all" season comparison of salinity values for the
entire time series: Generally, the salinities in winter and spring are highly significantly
lower compared to summer and autumn. Additionally, the summers show slightly
significantly higher salinities compared to autumn data.



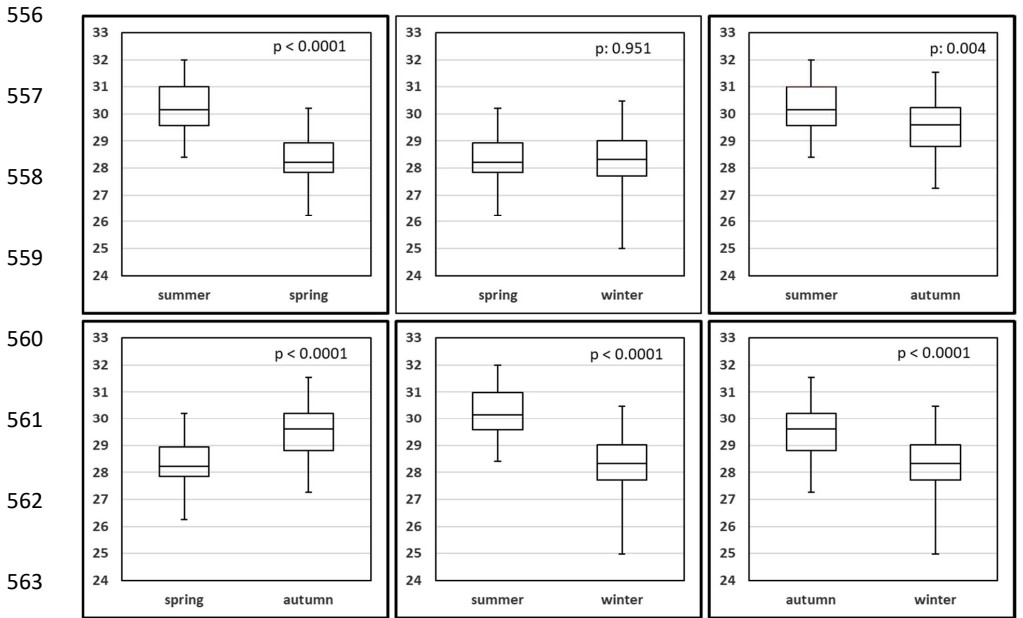

Figure 12: All-season comparison of salinity values for the entire time series.

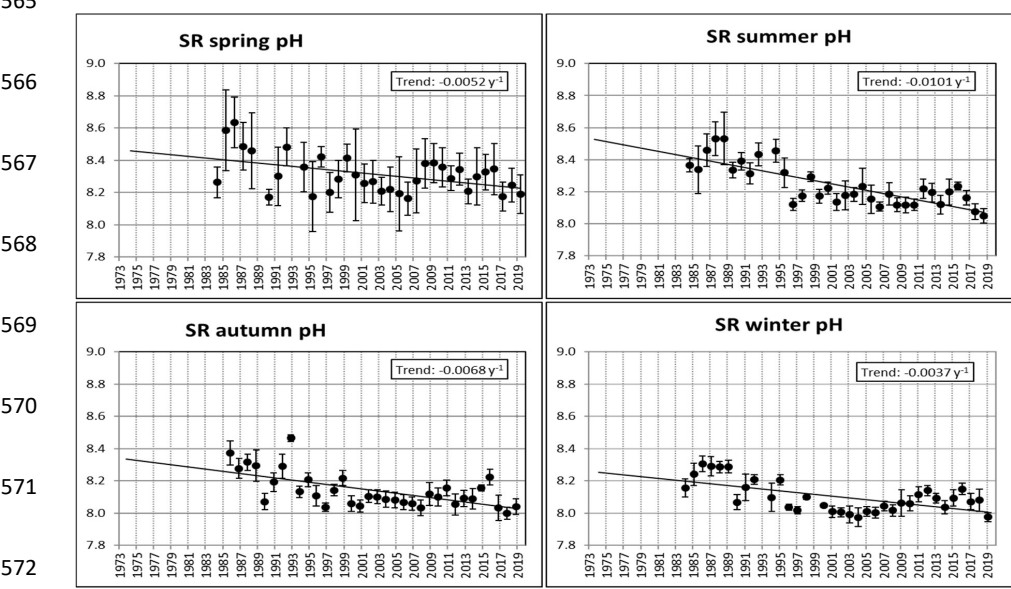

Figure 13a: pH development over the course of the Sylt Roads LTER time series. Seasonal averages with standard error of means (SEM) as error bars. Data on linear seasonal trends are shown in the boxes.



This overall picture is explained by the more prominent freshwater impact in winter
and spring to the area (Pätsch & Lenhart, 2004; van Beusekom et al., 2017).
Comparisons of seasonal salinities for the high and low eutrophication periods
yielded in no significant differences at all (Table A1 e).












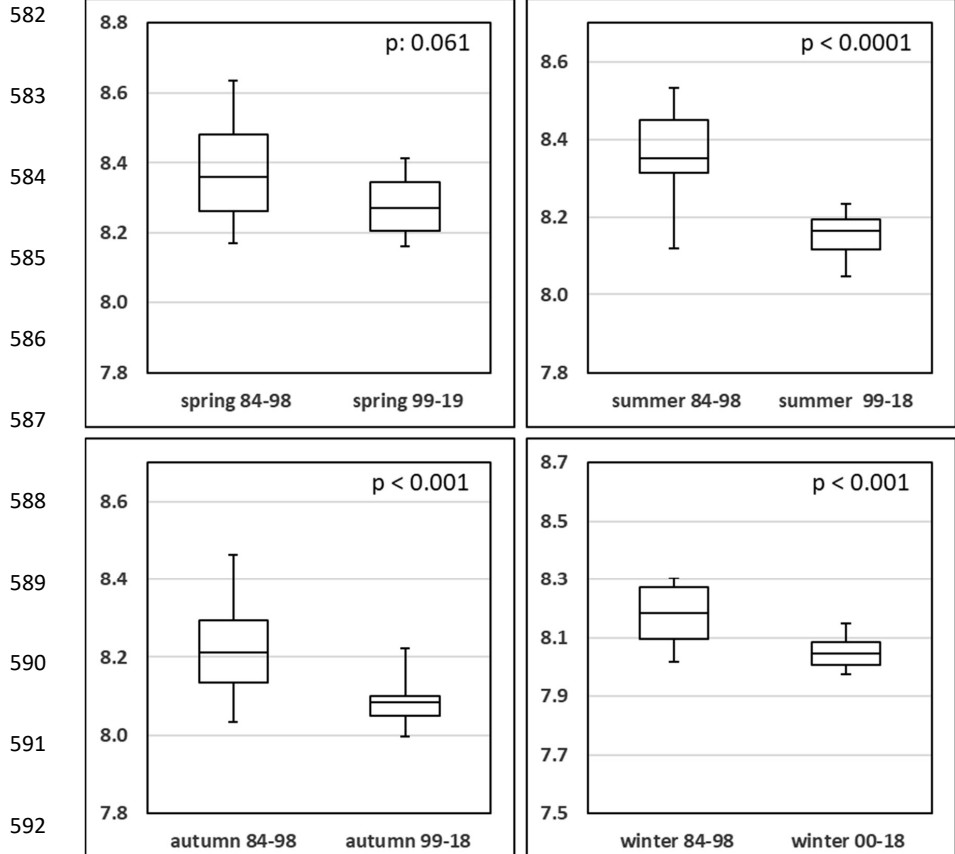

Figure 13b: Seasonal comparison of pH values for high/low eutrophication periods.

On average, the pH decreased since the start of continuous measurements in 1984
till 2019 by 0.23 units. This was evident for all seasons (Figures 13a) with summer,
autumn and spring showing most pronounced declines (-0.36, -0.24, -0.18). A t-test



comparing pH values from 1984-1998 with values from 1999-2019 yielded significant
differences for winter, summer and autumn seasons (p<0.001, Figure 13b, Table A1
g). Progressively declining pH levels in coastal regions have been documented
elsewhere e.g. from the US East Coast (Waldbusser et al., 2011, Wallace et al.,

2014).


**6. Related Datasets**

Over the years several data sets closely related to this physical-hydrochemical time
series were compiled at the Sylt Marine Observatory:
1. The **Sylt Roads zooplankton time series** was initiated by Peter Martens.

Quantification of abundant zooplankton (> 50 species/groups) occurred

weekly from 1979 to 2011. For this time period all data (32 years) are

stored in the open access repository PANGAEA (e.g. Martens, 2007,

2012). Due to the retirement of the lead scientist the series is on hold since

2012. Zooplankton samples are still taken weekly and stored for further

analysis.

2. The **Sylt Roads quantitative microplankton time series** was started in

June 1992 by Wolfgang Hickel. Mostly on a twice a week basis

microplankton abundance and related biomass parameters, such as

plankton biovolume and carbon were recorded. All data until 2013 are

compiled in the PANGAEA repository (Rick et al., 2017a)

3. In 1987, the **Sylt Roads semiquantitative microplankton time series**

was initiated by Gerhard Drebes, Malte Elbrächter and Hannelore Halliger.

Weekly in depth microscopic and regular electron microscopic analyses of



living plankton and fixed, respectively, samples resulted in high quality data
sets (> 700 taxa) compiled in PANGAEA until 2020 (Rick et al., 2018;
Castillo-Ramírez et al., 2021)
4.  In 1994, the **planktonic primary productivity and respiration time**
**series** was started by Ragnhild Asmus. Monthly measurements based on
the oxygen method (Gaarder and Gran, 1927) using oxygen sensitive
electrodes (WTW OxyCal) are ongoing in the List Königshafen area. All
data including 2014 are archived in PANGAEA (Asmus & Hussel, 2010;
Asmus, R., 2016a)
5.  The **Sylt Roads gelatinous zooplankton time series** was initiated by
Ragnhild Asmus. The data are available on a weekly basis since May 2009
(Asmus, R. et al., 2017 a, b)
6.  The **Sylt Roads bivalve larvae time series** was established in 1995 by
Matthias Strasser (Strasser & Günther, 2001). Twice a week sampling is
ongoing and the data are currently available via PANGAEA until 2014 (e.g.
Asmus, R., 2010, Asmus & Asmus, 2016)
7.  The **Sylt Roads Meroplankton time series** was established in 1996 by
Ragnhild Asmus. Sampling (twice a week) is ongoing and the data were
submitted to Pangaea in March 2022 (e.g. Kadel et al., submitted)
8.  The **Sylt Roads fish survey** was established in 2007 by Harald Asmus in
order to analyze the Wadden Sea fish fauna with special focus on
migration changes, species composition and feeding habits. Seven stations
are sampled monthly inside the Bight while two additional reference
stations, one outside the Bight and one close to the Danish border, are
sampled four times a year. The data are stored in the PANGAEA repository
from 2007 until 2020 (Asmus, H.  et al., 2020)



6. Data Access
Data retrieval is ensured via PANGAEA (Rick et al. (2017b-e, 2020a-o & Rick et al.,
submitted; doi:10.1594/PANGAEA.150032, 873549, 873545, 873547, 918018,

651    918032, 918027, 918023, 918033, 918028, 918024, 918034, 918029, 918025,

652    918035, 918030, 918026, 918036, 918031).



Open Access — Earth System Science Data Discussions

7. Appendix

| a. SRP | min | Q1 | median | Q3 | max | average | SEM | SD | variance | n | p | remarks |
|---|---|---|---|---|---|---|---|---|---|---|---|---|
| Spring 84-98 | 0.670 | 1.249 | 1.541 | 1.895 | 2.529 | 1.583 | 0.413 | 0.511 | 0.261 | 14 | 2.18 E-07 | HSD |
| 99-19 | 0.077 | 0.145 | 0.205 | 0.338 | 0.513 | 0.250 | 0.101 | 0.121 | 0.015 | 21 | | |
| Summer 84-98 | 0.582 | 0.795 | 0.873 | 1.286 | 3.585 | 1.185 | 0.456 | 0.708 | 0.501 | 15 | 0.0003 | HSD |
| 99-19 | 0.120 | 0.194 | 0.252 | 0.293 | 0.497 | 0.243 | 0.068 | 0.089 | 0.008 | 20 | | |
| Autumn 84-98 | 0.264 | 0.388 | 0.484 | 0.601 | 1.317 | 0.513 | 0.210 | 0.280 | 0.079 | 14 | 1.0 E-6 | HSD |
| 99-19 | 0.046 | 0.091 | 0.131 | 0.170 | 0.372 | 0.133 | 0.053 | 0.072 | 0.005 | 20 | | |
| Winter 84-98 | 2.111 | 2.457 | 2.578 | 3.036 | 3.913 | 2.755 | 0.412 | 0.487 | 0.237 | 15 | 1.1 E-10 | HSD |
| 00-19 | 0.618 | 0.741 | 0.873 | 1.056 | 1.336 | 0.866 | 0.157 | 0.189 | 0.036 | 20 | | |

| b. DIN | min | Q1 | median | Q3 | max | average | SEM | SD | variance | n | p | remarks |
|---|---|---|---|---|---|---|---|---|---|---|---|---|
| Spring 84-98 | 16.616 | 27.897 | 42.565 | 48.694 | 122.825 | 44.042 | 15.528 | 24.888 | 619.434 | 14 | 0.017 | SD |
| 99-19 | 11.094 | 17.817 | 22.847 | 28.455 | 44.126 | 25.246 | 7.261 | 8.855 | 78.413 | 21 | | |
| Summer 84-98 | 1.049 | 1.943 | 2.676 | 3.314 | 5.641 | 2.834 | 0.935 | 1.218 | 1.484 | 15 | 0.079 | MSD |
| 99-19 | 0.566 | 1.204 | 1.756 | 2.146 | 58.325 | 1.787 | 0.842 | 1.194 | 1.425 | 20 | | |
| Autumn 84-98 | 3.010 | 7.945 | 9.339 | 13.889 | 16.655 | 10.202 | 3.561 | 4.193 | 17.584 | 14 | 0.584 | NSD |
| 99-18 | 3.513 | 7.391 | 9.518 | 11.234 | 15.508 | 9.081 | 2.642 | 3.310 | 10.954 | 20 | | |
| Winter 84-98 | 25.271 | 46.666 | 51.586 | 61.329 | 82.092 | 51.623 | 11.646 | 15.309 | 234.367 | 15 | 0.010 | SD |
| 00-18 | 28.448 | 38.568 | 42.246 | 44.744 | 50.021 | 41.256 | 4.463 | 5.712 | 32.623 | 20 | | |

| c. Si | min | Q1 | median | Q3 | max | average | SEM | SD | variance | n | p | remarks |
|---|---|---|---|---|---|---|---|---|---|---|---|---|
| Spring 84-98 | 0.163 | 6.603 | 7.933 | 13.567 | 16.226 | 8.501 | 3.896 | 4.691 | 22.002 | 14 | 0.718 | NSD |
| 99-19 | 1.962 | 5.908 | 8.451 | 12.011 | 19.196 | 9.943 | 3.959 | 4.937 | 24.375 | 21 | | |
| Summer 84-98 | 1.434 | 2.536 | 3.156 | 4.236 | 8.480 | 3.449 | 1.273 | 1.709 | 2.919 | 15 | 0.001 | HSD |
| 99-19 | 0.962 | 1.282 | 1.616 | 2.015 | 3.060 | 1.677 | 0.489 | 0.606 | 0.368 | 20 | | |
| Autumn 84-98 | 0.369 | 2.191 | 4.268 | 5.150 | 7.035 | 3.874 | 1.558 | 1.849 | 3.420 | 14 | 0.026 | SD |
| 99-19 | 0.965 | 3.626 | 6.134 | 7.276 | 9.940 | 5.588 | 2.113 | 2.447 | 5.988 | 20 | | |
| Winter 84-98 | 13.185 | 19.282 | 20.635 | 24.317 | 32.853 | 21.880 | 3.843 | 4.802 | 23.060 | 15 | 1.16 E-06 | HSD |
| 99-19 | 16.742 | 31.893 | 33.473 | 34.832 | 40.717 | 32.980 | 3.717 | 5.552 | 30.828 | 20 | | |

| d. SST | min | Q1 | median | Q3 | max | average | SEM | SD | variance | n | p | remarks |
|---|---|---|---|---|---|---|---|---|---|---|---|---|
| Spring 84-98 | 5.55 | 7.12 | 7.53 | 8.35 | 9.78 | 7.73 | 0.77 | 0.99 | 0.99 | 15 | 0.104 | NSD |




| | min | Q1 | median | Q3 | max | average | SEM | SD | variance | n | p | remarks |
|---|---|---|---|---|---|---|---|---|---|---|---|---|
| 99-19 | 5.99 | 7.60 | 8.48 | 9.06 | 10.05 | 8.17 | 0.87 | 1.08 | 1.16 | 20 | 0.043 | SD |
| Summer 84-98 | 16.00 | 16.47 | 17.06 | 17.64 | 19.01 | 17.23 | 0.69 | 0.82 | 0.68 | 15 | | |
| 99-19 | 16.44 | 17.03 | 17.71 | 18.29 | 19.15 | 17.79 | 0.61 | 0.72 | 0.52 | 20 | | |
| Autumn 84-98 | 8.846 | 10.156 | 10.425 | 10.831 | 12.755 | 10.614 | 0.618 | 0.865 | 0.748 | 15 | 0.00036 | HSD |
| 99-19 | 10.745 | 11.140 | 11.610 | 12.116 | 13.473 | 11.781 | 0.605 | 0.765 | 0.585 | 20 | | |
| Winter 84-98 | -0.736 | 2.416 | 3.527 | 4.220 | 5.141 | 3.094 | 1.253 | 1.557 | 2.423 | 15 | 0.994 | NSD |
| 99-18 | 0.422 | 2.438 | 3.548 | 3.973 | 5.211 | 3.139 | 0.987 | 1.213 | 1.472 | 20 | | |
| | | | | | | | | | | | | |
| **e. Sal (1)** | min | Q1 | median | Q3 | max | average | SEM | SD | variance | n | p | remarks |
| Spring 84-98 | 26.244 | 28.053 | 28.903 | 29.248 | 30.222 | 28.535 | 0.832 | 1.011 | 1.021 | 15 | 0.136 | NSD |
| 99-19 | 26.399 | 27.824 | 28.210 | 28.377 | 29.476 | 28.106 | 0.466 | 0.640 | 0.410 | 21 | | |
| Summer 84-98 | 28.408 | 29.794 | 30.926 | 31.670 | 31.996 | 30.528 | 0.987 | 1.121 | 1.258 | 15 | 0.140 | NSD |
| 99-19 | 28.727 | 29.591 | 30.052 | 30.699 | 31.274 | 30.127 | 0.519 | 0.639 | 0.408 | 20 | | |
| Autumn 84-98 | 27.261 | 29.110 | 29.645 | 30.838 | 31.532 | 29.804 | 1.052 | 1.258 | 1.582 | 13 | 0.443 | NSD |
| 99-18 | 27.959 | 28.758 | 29.449 | 30.019 | 31.475 | 29.407 | 0.804 | 0.945 | 0.893 | 20 | | |
| Winter 84-98 | 24.989 | 27.677 | 28.450 | 29.605 | 30.469 | 28.532 | 1.006 | 1.316 | 1.731 | 15 | 0.433 | NSD |
| 99-18 | 26.584 | 27.763 | 28.284 | 28.585 | 29.860 | 28.213 | 0.661 | 0.817 | 0.668 | 21 | | |
| | | | | | | | | | | | | |
| **f. Sal (2)** | min | Q1 | median | Q3 | max | average | SEM | SD | variance | n | p | remarks |
| Autumn | 26.244 | 27.838 | 28.226 | 28.943 | 30.222 | 28.284 | 0.640 | 0.848 | 0.719 | 33 | 1.48 E-06 | HSD |
| Spring | 27.261 | 28.810 | 29.606 | 30.226 | 31.532 | 29.567 | 0.900 | 1.091 | 1.191 | 36 | | |
| Summer | 28.408 | 29.584 | 30.160 | 30.993 | 31.996 | 30.312 | 0.770 | 0.915 | 0.837 | 35 | 0,004 | HSD |
| Autumn | 27.261 | 28.810 | 29.606 | 30.226 | 31.532 | 29.567 | 0.900 | 1.091 | 1.191 | 33 | | |
| Summer | 28.408 | 29.584 | 30.160 | 30.993 | 31.996 | 30.312 | 0.770 | 0.915 | 0.837 | 35 | 3.41 E-14 | HSD |
| Spring | 26.244 | 27.838 | 28.226 | 28.943 | 30.222 | 28.284 | 0.640 | 0.848 | 0.719 | 36 | | |
| Summer | 28.408 | 29.584 | 30.160 | 30.993 | 31.996 | 30.312 | 0.770 | 0.915 | 0.837 | 35 | 3.61 E-12 | HSD |
| Winter | 24.989 | 27.713 | 28.330 | 29.024 | 30.469 | 28.298 | 0.801 | 1.065 | 1.135 | 36 | | |
| Autumn | 27.261 | 28.810 | 29.606 | 30.226 | 31.532 | 29.567 | 0.900 | 1.091 | 1.191 | 33 | 9.1 E-06 | HSD |
| Winter | 24.989 | 27.713 | 28.330 | 29.024 | 30.469 | 28.298 | 0.801 | 1.065 | 1.135 | 36 | | |
| Winter | 24.989 | 27.713 | 28.330 | 29.024 | 30.469 | 24.989 | 28.330 | 29.024 | 30.469 | 36 | 0.95100 | NSD |
| Spring | 26.244 | 27.838 | 28.226 | 28.943 | 30.222 | 26.244 | 28.226 | 28.943 | 30.222 | 36 | | |





| g. pH | min | Q1 | median | Q3 | max | average | SEM | SD | variance | n | p | remarks |
|---|---|---|---|---|---|---|---|---|---|---|---|---|
| Spring 84-98 | 8.170 | 8.263 | 8.359 | 8.482 | 8.635 | 8.380 | 0.130 | 0.148 | 0.022 | 13 | 0.060 | MSD |
| 99-19 | 8.162 | 8.208 | 8.273 | 8.345 | 8.413 | 8.272 | 0.063 | 0.073 | 0.005 | 21 | | |
| Summer 84-98 | 8.120 | 8.314 | 8.351 | 8.449 | 8.532 | 8.361 | 0.091 | 0.115 | 0.013 | 14 | 0.00002 | HSD |
| 99-19 | 8.049 | 8.118 | 8.166 | 8.195 | 8.234 | 8.158 | 0.043 | 0.051 | 0.003 | 20 | | |
| Autumn 84-98 | 8.035 | 8.134 | 8.211 | 8.294 | 8.465 | 8.211 | 0.097 | 0.117 | 0.014 | 14 | 0.0009 | HSD |
| 99-19 | 7.998 | 8.051 | 8.083 | 8.101 | 8.221 | 8.085 | 0.038 | 0.050 | 0.003 | 20 | | |
| Winter 84-98 | 8.016 | 8.097 | 8.182 | 8.273 | 8.305 | 8.176 | 0.085 | 0.097 | 0.009 | 14 | 0.0004 | HSD |
| 00-19 | 7.974 | 8.007 | 8.044 | 8.083 | 8.149 | 8.048 | 0.043 | 0.050 | 0.003 | 20 | | |

| h. SPM | min | Q1 | median | Q3 | max | average | SEM | SD | variance | n | p | remarks |
|---|---|---|---|---|---|---|---|---|---|---|---|---|
| Spring 84-92 | 20.448 | 24.721 | 27.551 | 36.204 | 61.743 | 33.677 | 10.077 | 12.809 | 164.071 | 8 | 0.015 | SD |
| 00-19 | 9.486 | 13.019 | 16.943 | 20.631 | 26.581 | 16.936 | 4.054 | 4.770 | 22.753 | 20 | | |
| Summer 84-92 | 11.343 | 16.153 | 16.437 | 16.682 | 25.519 | 16.901 | 2.381 | 3.661 | 13.406 | 9 | 2.0 E-5 | HSD |
| 00-19 | 3.624 | 4.614 | 6.295 | 6.894 | 8.628 | 6.117 | 1.226 | 1.454 | 2.115 | 20 | | |
| Autumn 84-92 | 5.407 | 7.540 | 7.908 | 9.225 | 12.996 | 8.126 | 1.988 | 2.435 | 5.927 | 8 | 0.243 | NSD |
| 99-18 | 2.347 | 5.613 | 7.847 | 8.877 | 13.275 | 7.422 | 2.198 | 2.773 | 7.690 | 20 | | |
| Winter 84-92 | 20.339 | 37.672 | 44.247 | 49.250 | 65.783 | 40.977 | 8.916 | 11.777 | 138.687 | 9 | 0.007 | SD |
| 99-19 | 18.897 | 24.425 | 27.762 | 31.819 | 44.512 | 28.515 | 5.342 | 6.613 | 43.738 | 20 | | |

| i. Chl a | min | Q1 | median | Q3 | max | average | SEM | SD | variance | n | p | remarks |
|---|---|---|---|---|---|---|---|---|---|---|---|---|
| Spring 84-98 | 6.683 | 8.525 | 10.625 | 16.578 | 41.797 | 14.911 | 6.199 | 8.847 | 78.268 | 14 | 0.175 | NSD |
| 99-19 | 3.300 | 9.085 | 10.314 | 12.194 | 19.795 | 11.060 | 2.287 | 3.205 | 10.274 | 21 | | |
| Summer 84-98 | 3.461 | 4.229 | 5.233 | 6.946 | 10.493 | 6.042 | 1.859 | 2.200 | 4.839 | 15 | 0.390 | NSD |
| 99-19 | 3.523 | 4.226 | 4.867 | 5.426 | 6.913 | 5.286 | 1.250 | 2.248 | 5.055 | 21 | | |
| Autumn 84-98 | 1.163 | 1.302 | 1.513 | 2.340 | 3.740 | 1.810 | 0.629 | 0.759 | 0.577 | 14 | 0.099 | NSD |
| 99-18 | 0.755 | 1.165 | 1.465 | 1.774 | 2.321 | 1.488 | 0.366 | 0.435 | 0.189 | 20 | | |
| Winter 84-98 | 1.276 | 3.313 | 4.320 | 5.003 | 5.753 | 3.911 | 1.175 | 1.379 | 1.903 | 15 | 0.314 | NSD |
| 99-19 | 2.622 | 3.035 | 3.582 | 3.937 | 5.261 | 3.625 | 0.502 | 0.627 | 0.394 | 20 | | |

| j. DIN/SRP | min | Q1 | median | Q3 | max | average | SEM | SD | variance | n | p | remarks |
|---|---|---|---|---|---|---|---|---|---|---|---|---|






Open Access — Earth System Science Data Discussions

| (continued) | min | Q1 | median | Q3 | max | average | SEM | SD | variance | n | p | remarks |
|---|---|---|---|---|---|---|---|---|---|---|---|---|
| Spring 84-98 | 10.231 | 23.279 | 36.167 | 54.111 | 206.208 | 51.74 | 29.90 | 47.21 | 2229.11 | 14 | 2.8 E-5 | HSD |
| 99-19 | 19.640 | 157.680 | 264.042 | 424.961 | 964.970 | 343.38 | 188.76 | 238.16 | 56720.93 | 21 | | |
| Summer 84-98 | 1.670 | 3.059 | 5.059 | 9.899 | 22.039 | 7.984 | 4.872 | 6.009 | 36.114 | 15 | 0.007 | SD |
| 99-19 | 6.901 | 6.901 | 12.770 | 28.299 | 65.835 | 20.174 | 14.832 | 17.846 | 318.492 | 20 | | |
| Fall 84-98 | 3.235 | 4.614 | 5.661 | 7.593 | 8.806 | 5.558 | 1.719 | 1.992 | 3.970 | 14 | 2.4 E-6 | HSD |
| 99-18 | 2.393 | 9.018 | 10.794 | 14.957 | 23.639 | 12.167 | 3.227 | 3.981 | 15.845 | 20 | | |
| Winter 84-98 | 5.985 | 20.789 | 24.408 | 27.149 | 30.496 | 22.953 | 4.496 | 5.545 | 30.749 | 15 | 1.0 E-10 | HSD |
| 99-19 | 11.838 | 44.347 | 48.459 | 54.687 | 79.728 | 51.408 | 7.747 | 10.510 | 110.455 | 20 | | |

| k. DIN/Si | min | Q1 | median | Q3 | max | average | SEM | SD | variance | n | p | remarks |
|---|---|---|---|---|---|---|---|---|---|---|---|---|
| Spring 84-98 | 4.656 | 5.699 | 17.727 | 89.838 | 935.061 | 158.576 | 197.078 | 264.416 | 69915.647 | 14 | 0.110 | NSD |
| 99-19 | 0.517 | 11.661 | 20.777 | 28.856 | 50.656 | 22.338 | 9.369 | 11.782 | 138.826 | 21 | | |
| Summer 84-98 | 0.269 | 0.856 | 1.949 | 8.357 | 97.664 | 15.361 | 18.789 | 26.587 | 706.861 | 15 | 0.102 | NSD |
| 99-19 | 0.525 | 1.263 | 1.449 | 1.958 | 5.616 | 1.930 | 0.915 | 1.274 | 1.622 | 20 | | |
| Autumn 84-98 | 0.382 | 0.704 | 1.039 | 1.139 | 185.168 | 15.208 | 24.424 | 47.420 | 2248.656 | 14 | 0.325 | NSD |
| 99-18 | 0.345 | 0.422 | 0.503 | 0.623 | 2.599 | 0.744 | 0.411 | 0.630 | 0.397 | 20 | | |
| Winter 84-98 | 1.324 | 2.254 | 2.624 | 3.213 | 14.752 | 3.754 | 2.047 | 3.303 | 10.912 | 15 | 0.018 | SD |
| 99-19 | 1.112 | 1.276 | 1.339 | 1.374 | 1.993 | 1.355 | 0.103 | 0.175 | 0.031 | 20 | | |

| l. n | min | Q1 | median | Q3 | max | average | SEM | SD | variance | n | p | remarks |
|---|---|---|---|---|---|---|---|---|---|---|---|---|
| Spring 74-98 | 12 | 17.25 | 26 | 38.75 | 77 | 29.762 | 14.165 | 17.635 | 310.994 | 22 | 0.219 | NSD |
| 99-19 | 6 | 20 | 24 | 34 | 38 | 26.4 | 6.925 | 8.08 | 65.293 | 21 | | |
| Summer 73-98 | 11 | 26.5 | 40 | 51.5 | 198 | 48.682 | 21.928 | 36.463 | 1.329.584 | 23 | 0.0197 | SD |
| 99-19 | 22 | 24 | 26 | 33 | 39 | 28.75 | 5.17 | 5.893 | 34.726 | 21 | | |
| Autumn 73-98 | 13 | 22 | 29 | 45 | 80 | 34.45 | 15.084 | 18.743 | 351.283 | 21 | 0.088 | NSD |
| 99-18 | 13 | 23 | 25 | 31.75 | 41 | 27.421 | 6.14 | 7.379 | 54.448 | 20 | | |
| Winter 73-98 | 6 | 14.75 | 27.5 | 38.75 | 50 | 26.857 | 11.273 | 13.525 | 182.926 | 22 | 0.05 | SD |
| 99-19 | 12 | 18 | 19 | 25 | 32 | 21.65 | 4.694 | 5.333 | 28.440 | 21 | | |

Table A1: Descriptive statistics related to boxplot figures (4b, 5-10, 11b, 13b) with p-values of associated t-tests (two sided, unequal variances assumed) comparing seasonal data for two time periods within the Sylt Roads LTER characterized by different eutrophication potential (High: 1978-1998; Low: 1999-2019). In case of salinity (part e. of Table) seasons are compared to each other for the complete series (1973-2019). Q1 = 1st quartile; Q3 = 3rd quartile; SEM: standard error of means; SD: standard deviation



8. Author contribution
JR prepared the manuscript with the contribution of the following co-authors (MS, TR,
JB, RA, HA, FM, AK, KW). RS compiled the data in Pangaea. TR performed the
hydrochemical measurements since 2000.
9. Competing interest
The authors declare that they have no conflict of interests.
10. Acknowledgements
We would like to thank all the devoted people who have been supporting the Sylt
Roads LTER time series over more than four decades. Special thanks go to Ludmila
Baumann, Tanja Burgmer, Lydia Canals, Marthe Claußen, Gerhard Drebes, Claus-
Dieter Dürselen, Malte Elbrächter, Peter Elvert, Alfonso Lopez Gonzales, Alexandra
Halbe, Hannelore Halliger, Wolfgang Hickel, Valentin Hildebrand, Birgit Hussel, Petra
Kadel, Alexandra Kraberg, Niels Kruse, Peter Martens, Cornelia Reineke, Karsten
Reise, Alfred Resch, Anette Tillmann and Kay von Böhlen.










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
