# Peer review of "An evaluation of long-term physical and hydrochemical measurements at the Sylt"

_Earth System Science Data, 2020_

## Author Response (AR1)

Dear Dr. Fleischer,

Please find in this letter a detailed point by point response to the comments of the reviewers. This is structured in the following way:

First, we show the original comments of the reviewers. Second, you will find in brackets [**bold**] our suggestions submitted to you on December 5[th]. Third, we explain the corresponding changes [[*Italic*]] in the revised manuscript by specifying the line numbers involved.

reviewer 1:

The paper "An evaluation of long-term physical and hydro chemical measurements at the Sylt Roads Marine Observatory (1973-2019), Wadden Sea, North Sea" by Rick et al covers the time series data for STT, salinity and nutrients. This manuscript/datasheet presents great information and long-term trends in of hydrochemical parameters which are of interest to larger biogeochemistry community. I believe this manuscript contributes to our understanding and observation of changes occurring in the physical parameter and influences in the nutrient dynamics. The manuscript is acceptable in the current version, however address few points below can make the manuscript better.

1. Figure 2a doesn't add any significant information and removing this figure can make the paper compact.

   [**Figure 2a will be removed and the information will be given as text only.**]

   [[ *The original Figure 2 a was removed from the manuscript. The information of the figure is given in the text (lines 146 – 154), marked yellow. Additionally, the former Fig. 2b was renamed Fig. 2, Line 170*]]

2. The overall analysis/discussion of the physical and hydrochemical measurement can help to better quantify the relation among them. Such as PCA or any other statistical analysis would make the manuscript robust.

   [**We will add a PCA analysis based on seasonal averages of the parameters.**]

   [[*We added a correlation and a principal component analysis of the data (seasonal averaged) to the manuscript (lines 333 – 389). The new table 3a (lines 349-352) shows the results of the correlation analysis. The likewise new table 3b (lines 360-363, Variance Table) shows the outcome of the PCA. A PCA Correlation Monoplot is shown in the new Figure 4 (lines 376-379). All following figures were renamed accordingly and are highlighted with yellow in the manuscript.*

   *In the Appendix the Table A2 on the PCA coefficients was added displaying the quantitative importance of the individual parameters in each principal component (lines 727-739)*]]

reviewer 2:

Dear authors,

The paper you have submitted is a valuable addition to the future research on global change and possible effect detection. The datasets are spanning a long time period and in combination with other datasets from the German bight it becomes tremendously valuable to other researchers.

Nevertheless, there are some aspects which came to my attention trying to review the datasets.

The figure 2/a) seams trivial and with only little additional value to the existing text

[**As stated already the figure 2a will be removed.**]

[[ *Figure 2a was removed.]]*

The figure 2/b) are the inserts really necessary? The overall information about more standardized sampling efforts in the later years seams obvious from the scatter plots already. I would argue, that it would be possible to combine all 4 seasons, by different shapes and colors and downscaling the size of the figure. It remains that it is valuable information to see the sampling effort over time.

[ **We would prefer to keep the figure as is, but if you as an editor would like to go with the suggestion of reviewer 2 (compress the four subfigures to only one) we will change the figure accordingly**.]

[[*We kept Fig 2a, but renamed it to Fig 2. But as already said, we will change it according reviewer 2, if desired*.]]

In the text of this page it seems that there is some confusion in the text. Lines 157 "more than 63000 data were collected..." in a data journal I expect more precise information on what was collected. Suggestion: "data points". In the same line, it is mentioned, that in 5700 RV Mya Cruises? 5700 cruises would mean 15 years of 1 cruise per day. I'm guessing, that there is confusion between "Events" and cruises. One cruise contains several sampling events, so maybe there is a misinterpretation. Especially, in comparison with the 300 land base samplings, this number sounds more reasonable.

[ **We will put in the exact number of data points as well as the exact numbers of ship- and land-based samplings.**]

[[*We checked the data and put the exact numbers in (lines 149-154).*]]

The detailed figures you present are difficult to compare, because the combination of lines and datapoints in the same figure for such a long time period make the figures difficult to read and, in some areas, just a black bubble. This is also increased by sticking to the x-axis scaling always to 1973, instead of spreading the shorter timeseries in the figures for better readability. A comment in the caption could be information enough to the reader.

**[Well, the single data points were shown in the first place to underline the fact that the peaks are based on several but just one measurement - but I see the point that to many dots sometimes may blur the information. Accordingly, the data will be displayed as line charts only. On the other hand, we strongly discourage the suggestion to adjust the time axis case related in every plot. Since a lot of parameters have different "starting times "(1973, 1979, 1984) this would enhance the possibility of misinterpretation by the reader. Additionally, a case related shortening/adjusting of the time axis (e.g. 1973 - 2019 to 1979 - 2019) would not enhance readability in a significant manner.]**

[[*We changed all the subfigures (3a-3j) of Figure 3 from dot line presentation to only line presentation, but kept the original time period 1973-2019 on the x-axis. This way the graphs appear less "blurry" in the "low concentration" range and additionally allow for a good comparability between the subfigures*.]]

Example SST 3/b: May be plotting the summer values and the winter values (or annual max and min values) to shorten the range could be a better description of what to expect when using this dataset. I would like to see more combined figures of the different stations, instead of the Sylt Roads alone. But a good visualization is key to this.

[**Please see our statement above.**]

[[ *Seasonal values of SRP are given in Figure 5a as an example. Additionally, the boxplot in Figures 5b – 8 give a good overview on the seasonal variability (including max and min values) of the parameters*.]]

The section related datasets is far too large, the size of the paper should be more compact. In any case the usability of the data needs some more investment. The years of 2014 to 2019 are directly available and require several downloads. But for the earlier years it is necessary to follow the DOI: https://doi.org/10.1594/PANGAEA.150032 which then provides about 40 to 50 new DOIs which to download. Hopefully, there are no more of these collection DOIs included in these new DOIs. The cascading style of the PIDs is not very friendly to users. I know that this may be out of the control of the authors, but this is also a comment to the editors, to avoid such cascading PIDs for the future of reviewing the data.

[**Totally agreed! But as the reviewer already states it was unfortunately out of our control. I even prepared one data set covering all measurements of the time series and tried to submit it to PANGAEA but I was told that they won't accept it if the data are published in PANGAEA somewhere else ....]**

[[ *See above!*]]

Additional changes to the manuscript not related to any reviewer comment:

- We put in additional information on the statistical analyses (lines 138-142).
- We changed the information within 6. Related Datasets 4. "planktonic primary productivity", since this part of the time series came to an end in 2020 (lines 687-688).
- We changed the information within 6. Related Datasets 7 Sylt Roads Meroplankton time series. The data are now published in PANGAEA (lines 697-699).

- We added the corresponding publication (Strasser et al., 2022) to 10. References (lines 924-927).

I hope we were able to explain the changes to the manuscript in an understandable way.

I wish you a happy and relaxing holiday season!

Best regards,

Johannes Rick

---

## Author Response (AR2)

Dear Dr. Fleischer,

We refer to our response of Dec. 14th 2022.

Additionally, we now provide a clean version of the manuscript file as *.pdf.

We have merged the figures as follows:

Figures 3a-3j into one Figure 3 (a-j); Figures 5a and 5b into one Figure 5 (a, b); Figures 12a and 12b into one Figure 12 (a, b); Figures 14a and 14b into one Figure 14 (a, b).

We have merged the tables as follows:

Tables 3a and Tables 3b into one Table 3 (a, b).

Additionally, we noticed some slight mistakes and corrected them:

1. In Table 1: Changed one reference from "Grasshoff & Wenck (1983)" to "Grasshoff & Johannsen (1974)". Did the same in 10. References
2. Lines 537-541: We corrected and rephrased the text

**from**

"The spring and winter DIN/Si ratios (Figure 10, Table A1 k) moved from higher

(1973-1998) to more balanced values (1999-2019). In winter (p = 0.018) this change

is significant.  For the summer and autumn comparisons DIN/Si remained close to a

ratio of 1."

**to**

"The winter DIN/Si ratio (Figure 10, Table A1 k) changed significantly (p = 0.018)

from higher (1984-1998) to more balanced values (1999-2019).  For the summer and

autumn comparisons DIN/Si remained close to a ratio of 1. The comparison for the

spring seasons shows no significant change but is characterized by clearly lowered

min and max DIN/SRP values in more recent years."

3. We corrected some typos in section 6. Related datasets (e.g. line 726 from Asmus, R. et al., to Asmus et al..

Best wishes and a happy holiday season,

Johannes Rick